



# A probabilistic framework for quantifying the role of anthropogenic climate change in marine-terminating glacier retreats

John Erich Christian[1,2], Alexander A. Robel[2], and Ginny Catania[1]

[1]Institute for Geophysics, University of Texas at Austin
[2]School of Earth and Atmospheric Sciences, Georgia Institute of Technology

**Correspondence:** John Erich Christian (johnerich.christian@austin.utexas.edu)

**Abstract.** Many marine-terminating outlet glaciers have retreated rapidly in recent decades, but these changes have not been formally attributed to anthropogenic climate change. A key challenge for such an attribution assessment is that if glacier termini are sufficiently perturbed from bathymetric highs, ice-dynamic feedbacks can cause rapid retreat even without further climate forcing. In the presence of internal climate variability, attribution thus depends on understanding whether (or how frequently) these rapid retreats could be triggered by climatic noise alone. Our simulations with idealized glaciers show that in a noisy climate, rapid retreat is a stochastic phenomenon. We therefore propose a probabilistic approach to attribution and present a framework for analysis that uses ensembles of many simulations with independent realizations of random climate variability. Synthetic experiments show that century-scale climate trends substantially increase the likelihood of rapid glacier retreat. This effect depends on the timescales over which ice dynamics integrate forcing. For a population of synthetic glaciers with different topographies, we find that external trends increase the number of large retreats triggered within the population, offering a metric for regional attribution. Our analyses suggest that formal attribution studies are tractable and should be further pursued to clarify the human role in recent ice-sheet change. We emphasize that early-industrial-era constraints on glacier and climate state are likely to be crucial for such studies.

## 1 Introduction

Many marine-terminating glaciers in Greenland and Antarctica have retreated in recent decades, causing increased discharge to the ocean and accelerating sea-level rise (e.g., Meredith et al., 2019; Mouginot et al., 2019; Rignot et al., 2019). In Greenland, marine-terminating glacier retreat has been widespread across the ice sheet, although heterogeneous within regions (Howat et al., 2008; Moon and Joughin, 2008; Murray et al., 2015; King et al., 2018; Catania et al., 2018). In Antarctica, grounding line retreat and dynamic thinning is underway in the Amundsen Sea Embayment (ASE; e.g., Rignot et al., 2014; Joughin et al., 2014), and widespread retreat of glacier termini has also been documented on the Antarctic Peninsula (Cook et al., 2014) and parts of East Antarctica (Miles et al., 2013).

In both Greenland and Antarctica, glacier retreat has been linked to ocean forcing (Holland et al., 2008; Straneo et al., 2013; Jenkins et al., 2016; Wood et al., 2021), which in many cases is also linked to atmospheric changes. For example, wind anomalies in the Amundsen sea are thought to drive warm Circumpolar Deep Water towards ASE glaciers (e.g., Thoma et al.,





2008), while in Greenland, surface meltwater from atmospheric warming is discharged subglacially at glacier termini and affects heat exchange between the ice and fjord waters in buoyant plumes (e.g., Straneo et al., 2011; Fried et al., 2015). In many cases, these local atmospheric and ocean anomalies can be traced to large-scale modes of internal climate variability. Tropical Pacific variability has been tied to submarine melt for ASE glaciers (Steig et al., 2012; Dutrieux et al., 2014; Holland et al., 2019), and canonical modes of North Atlantic ocean and atmospheric variability have been concurrent with glacier retreat

in Greenland, especially in the Southern and central sectors of the ice sheet (e.g., Straneo et al., 2013; Khazendar et al., 2019). Though observations of glacier change are overwhelmingly concentrated in the last few decades, historical records and proxy evidence suggest links between these climate modes and glacier retreat earlier in the 20th century as well (e.g., Andresen et al., 2012; Bjørk et al., 2012; Andresen et al., 2014; Smith et al., 2017).

The sensitivity of marine-terminating glaciers to internal climate variability raises the question as to whether recently ob-

served retreats are caused by natural variability or anthropogenic climate change. Arctic and Antarctic land-surface warming has been formally attributed to anthropogenic forcing (Gillett et al., 2008), and recent studies have identified anthropogenic trends in additional climate variables relvant to glacier forcing, such as the winds that modulate ocean circulation near west-Antarctic glaciers (Holland et al., 2019; O'Connor et al., 2021). However, attribution of observed glacier retreats must incorporate the complex dynamics that link local climate to glacier response. Despite advances in understanding marine-terminating

glacier dynamics and ice-ocean interactions in recent decades (e.g. Jenkins et al., 2016; Catania et al., 2020), observed retreats and the associated dynamic mass loss have yet to be attributed to anthropogenic forcing, as detailed in the IPCC's Special Report on the Oceans and Cryosphere in a Changing Climate (SROCC; Meredith et al., 2019), and most recent assessment report (AR6; IPCC, 2021).

Why have marine-terminating glacier changes not yet been formally attributed? Strong variability in high-latitude climate,

multiple forcing processes, and limited observations are among the challenges reviewed in the SROCC and AR6. The dynamics of marine-terminating glacier flow and the potential for internal glacier instabilities pose additional challenges, but have not yet been addressed in formal attribution analyses. In this study we propose a way forward, developing a probabilistic attribution framework that accounts for marine-terminating glacier dynamics in the context of natural climate variability. Before presenting our methods and results, we briefly review the key elements of previous climate attribution analyses, and outline specific

challenges that we will consider for marine-terminating glacier retreats.

Attributing changes in any part of the Earth system to anthropogenic forcing is typically accomplished by comparing model simulations of that system with anthropogenic forcings included to simulations in which anthropogenic forcings are omitted. For example, climate model simulations including all major natural (e.g., volcanic and solar) and anthropogenic forcings (e.g., greenhouse gases and aerosols) are compared against counterfactual simulations with only the natural forcings (e.g., IPCC,

2021). Only the simulations including all known sources of forcing on the climate system match observed warming trends, giving high confidence that the global-mean warming over the industrial era is attributable to human influence, rather than natural forcings or internal variability. Another category of attribution studies focuses on extreme events, such as individual heat waves, floods, or storms (see e.g., Stott et al., 2016, for a review). Because extreme events would still occur due to internal variability in a natural climate, these studies focus on how anthropogenic forcing changes the likelihood of a given extreme.



While this framing is different from that of trend attribution, these studies still rely on comparing simulations with and without anthropogenic forcing. For example, Stott et al. (2004) examined temperature extremes in climate model output and concluded that European heat waves as severe as that of 2003 had become at least twice as likely due to anthropogenic forcing. Many studies on subsequent events have followed, and methods for event attribution have become increasingly standardized (Philip et al., 2020).

Attribution studies within glaciology have thus far focused on mountain glacier losses over the industrial era (Marzeion et al., 2014; Roe et al., 2017, 2021). These studies require models that accurately capture how glacier dynamics integrate both long-term trends and short-term climate variability, and have shown that the loss of mountain glaciers over the industrial era is clearly attributable to anthropogenic forcing, both for the metrics of terminus retreat (Roe et al., 2017) and mass balance (Roe et al., 2021). Indeed, the kilometer-scale retreats of mountain glaciers are some of the most statistically robust indicators of

anthropogenic climate change observed in the Earth system (Roe et al., 2017).

In all of these examples, attribution relies crucially on capturing the full scope of natural variability within a system. For marine-terminating glaciers, observations (discussed above), as well as theory and models (Robel et al., 2018; Christian et al., 2020), indicate that glacier termini fluctuate in response to stochastic climate variability. However, the largest observed retreats are overwhelmingly associated with retrograde bed slopes (i.e., deepening inland), both for marine-terminating glaciers in

fjords in Greenland (Catania et al., 2018; Wood et al., 2018) and for large outlet systems in West Antarctica, such as Pine Island and Thwaites glaciers (Rignot et al., 2014; Joughin et al., 2014). This implicates the marine ice sheet instability (e.g., Weertman, 1974; Schoof, 2007), where termini on retrograde beds may undergo runaway retreat if sufficiently perturbed, due to a positive feedback between ice flux and thickness at the terminus. Retreats continue until the inland topography becomes sufficiently shallow, or may lead to complete glacier retreat.

The cases where instabilities contribute to observed glacier-terminus retreats present a unique challenge for attribution studies, because they raise the possibility that natural variability could trigger a retreat of the same magnitude as one driven by anthropogenic forcing. In other words, large marine-terminating glacier retreats are not necessarily indicators of a significant climate change. Just as extreme events require a different attribution framework compared to other climate phenomena, retreats of marine-terminating glaciers may also require a unique framework when dynamic instabilities are possible. Our guiding

question in this study is thus, "what is the likelihood that natural variability alone drives rapid glacier terminus retreat, and how does an anthropogenic trend in forcing change this likelihood?" To address this question, we introduce an ensemble-based attribution framework (Section 2.3) as a tool for assessing the retreat of single glaciers and groups of glaciers. We note that this study is not itself a formal attribution assessment for a particular glacier or region. Rather, we identify key physical principles that affect the likelihood of rapid marine-terminating glacier retreats in a noisy climate, and show how model ensembles can

be used to clarify the effects of anthropogenic forcing. Our main goal is to put forward a probabilistic framework that can be used as a methodological guide for future attribution studies.

We first introduce the model and ensemble methods (Section 2). Second, we explore the null hypothesis of noise-triggered glacier retreat, and analyze controls on the likelihood of such retreat (Section 3). Third, we assess the effects of anthropogenic forcing and present a synthetic attribution experiment (Section 4). Our final set of results shows how to frame attribution for a



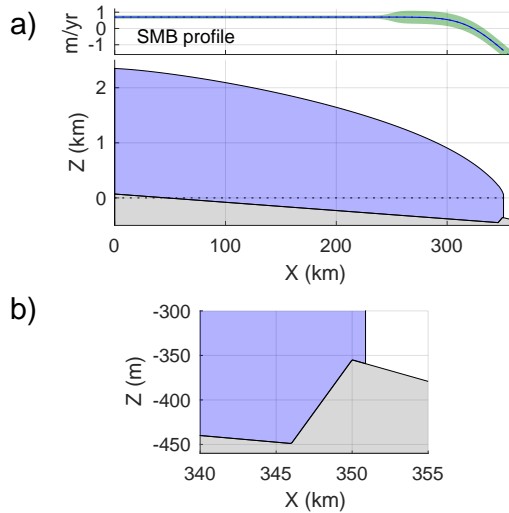

**Figure 1. (a)** An idealized marine-terminating glacier, whose steady-state terminus position is approximately 1 km from a bed peak. SMB profile is shown on top; shading shows the magnitude of variability ($1\sigma$). **(b)** Close-up of the bed peak.

population of glaciers (Section 5). We close with discussion of our findings, key uncertainties, and recommendations for how the community could proceed with formal attribution studies.

## 2 Methods

### 2.1 Glacier model and idealized geometry

We use a 1-D model of marine-terminating glacier dynamics that assumes flow is dominated by longitudinal stretching and Weertman-style sliding (see Supplement for equations). Ice velocities are calculated using the shallow shelf/stream approximation (SSA), ice thickness is calculated through mass conservation, and the terminus position is calculated using an implicit flotation condition. Following the numerical approach laid out in Schoof (2007), the governing equations are solved on a grid stretched to the glacier extent at each time step. That is, if the glacier advances or retreats, the grid nodes shift with respect to a fixed spatial coordinate. The model does not simulate any floating ice, so the last grid point always represents the grounding line (i.e., a terminus at flotation). The grid spacing is several kilometers through most of the interior, but is significantly refined ($\sim 100$ m) near the grounding line. Ice flux across the grounding line is not parameterized, but rather calculated directly from the local stress balance. Marine-terminating glacier models using this numerical approach have been benchmarked and used in many prior studies (e.g., Schoof, 2007; Robel et al., 2018).

The majority of the simulations in this study investigate an idealized glacier whose terminus position is near a small bed peak (Fig. 1). Our standard geometry has an inland prograde bed slope of $1.5 \times 10^{-3}$, and a sharp peak 350 km from the ice





divide. On its landward face, the peak rises 88–100 m (depending on experiment) in 4 km, and its seaward face has a slope of approximately $5 \times 10^{-3}$ (Fig. 1). We chose this idealized peak in order to provide a very distinct boundary between stable and unstable terminus positions. However, we introduce more realistic bed geometries with many bed peaks of varying geometry in Section 5. The time-averaged surface mass balance (SMB) is $+0.7$ m/yr through most of the interior and smoothly decreases

over the last $\sim 50$ km to approximately $-1.4$ m/yr at the terminus (Fig. 1a, top). We simulate ocean forcing very generally through a frontal ablation term at the grounding line (see Supplement for numerical implementation). Our standard glacier has a mean frontal ablation rate of 30 m/yr. This is within the range of observed submarine melt rates, which vary by orders of magnitude (c.f., Motyka et al., 2011; Mouginot et al., 2015). This simple frontal ablation scheme primarily serves as a localized flux term that can vary in time. For real glaciers, flux anomalies could be caused by anomalous calving or submarine melt, or

a combination of these processes.

## 2.2 Climate and glacier variability

Internal climate variability, which arises from chaotic fluctuations in atmospheric and ocean circulation, directly affects accumulation, surface melt, and ocean forcing on marine-terminating glaciers. We simulate the effects of internal climate variability on both SMB and frontal ablation as a first-order autoregressive process (AR-1), which is a common model for realistic climate

variability with temporal persistence (i.e., "memory"; Hasselmann, 1976). We generate synthetic variability using a Fourier transform method (see Percival et al., 2001; Roe and Baker, 2016; Christian et al., 2020). This allows us to generate random frontal ablation and SMB timeseries that are correlated with each other (i.e., reflecting the same regional climate anomalies) but which have different levels of prescribed persistence.

We set a decorrelation timescale of 10 years for frontal ablation anomalies, which emulates the decadal variability known

to be an important aspect of ocean forcing in both Greenland and Antarctica (e.g., Andresen et al., 2012; Straneo et al., 2013; Jenkins et al., 2016). We explore a range in the magnitude of frontal ablation variability across our experiments, with standard deviations from $\sigma_{FA} = 12$ m/yr to 27 m/yr (when sampled annually). We set a lower bound of zero on absolute frontal ablation values (i.e., we truncate anomalies more negative than 30 m/yr). This avoids numerical issues that can follow strongly negative ablation anomalies, which add ice to the terminus. We note that this lower bound introduces an asymmetry in the variability for

large $\sigma_{FA}$, which affects the mean state as variability is increased. However, based on tests without a zero bound, this appears secondary to the main effects of increasing overall variability (see Supplement, Fig. S2). It should also be noted that other nonlinearities in the ice dynamics can affect the mean state when variability is imposed (e.g., Robel et al., 2018).

For SMB, we set a decorrelation timescale of 1.5 years, consistent with higher-frequency atmospheric variability. We set $\sigma = 0.4$ m/yr near the terminus, tapering to $\sigma = 0.1$ m/yr inland, reflecting the greater variability in snowfall and melt near ice

sheet margins (e.g., Fyke et al., 2014).

Figure 2 shows a single realization of stochastic SMB and frontal ablation forcing (the latter with $\sigma_{FA} = 12$ m/yr) and the response from the idealized glacier shown in Fig. 1. The grounding line fluctuates in a roughly 4 km range on the seaward side of the bed peak for over 14 kyr before retreating permanently from the peak due to one particularly persistent (but random) anomaly in both frontal ablation and SMB. Retreat continues over several centuries until the terminus reaches a new stable



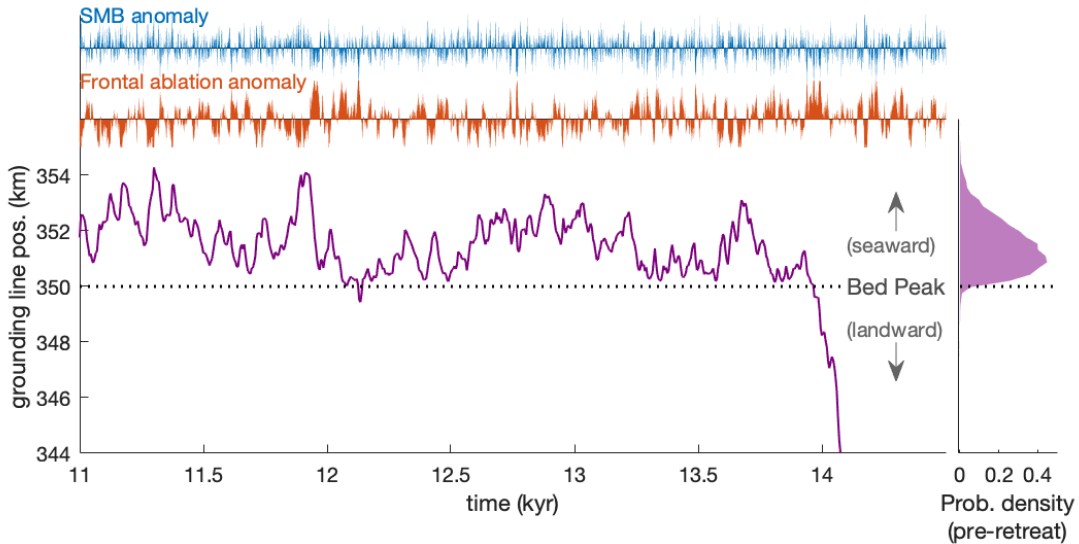

**Figure 2.** A synthetic glacier's response to stochastic variability, when the terminus is near a peak in bed topography. Normalized climate anomalies are shown in the upper axes, and grounding-line fluctuations are below. Dashed line indicates the location of the bed peak. A long period of stable, stationary fluctuations precedes unstable retreat off of the bed peak. The histogram (right) shows the probability distribution of stable fluctuations.

mean state several 10s of km inland (not shown), consistent with the marine ice sheet instability. This simulation illustrates the basic null hypothesis for attribution: a sustained retreat from a bed peak initiated purely by natural variability. It also highlights the stochastic nature of such retreats: the glacier fluctuates for a long time in close proximity to the topographic threshold, before permanently retreating. The frontal ablation and SMB anomalies that triggered retreat occurred purely by chance in the sequence of stochastic forcing.

## 2.3 Ensemble framework

The stochastic nature of noise-driven retreat from a bed peak (Fig. 2) suggests that for attribution, it may be more productive to consider such retreats as discrete events rather than trends; and to focus on their likelihood rather than their total magnitude, which is set largely by bed topography. Our method is to run large ensembles (i.e., collections) of simulations with the glacier model, where each simulation is forced with an independent realization of random climate variability drawn from the same distribution. This assumes that the statistics of the forcing variability are related to constant physical characteristics within the climate system, but the particular sequence of anomalies is fundamentally random. Because any realization of such climate variability is equally plausible, the ensemble thus provides a statistical population from which we can estimate the likelihood of a particular event, such as a sustained retreat from the bed peak. We refer to these ensembles as "aleatory" ensembles because the only difference between members is the realization of random forcing.





Unless otherwise indicated, we assess glacier retreats within an experimental interval of 150 years, corresponding to the approximate time frame of significant anthropogenic forcing (e.g., IPCC, 2021). Within this interval, we count ensemble members with termini that retreat more than 4 km behind the bed peak. This threshold corresponds to the inland extent of the retrograde bed (Fig. 1), so it is a reasonable indicator that the retreat is due to a loss of terminus stability and that the terminus will continue retreating (indeed, this is observed in longer simulations). However, since stability thresholds can be ambiguous

for transient glacier states (Sergienko and Wingham, 2021; Robel et al., 2021), we refer to retreats exceeding this threshold as rapid and sustained retreats. The threshold is within the range of many observations of retreats from bed peaks (e.g., Catania et al., 2018; Wood et al., 2018). For attribution assessments on real glaciers, the threshold for what counts as a retreat within ensemble simulations would be informed by the observed retreat extent. We return to the question of defining retreats in the discussion section.

The attribution problem we are focusing on assumes that a glacier's pre-industrial terminus position was near a bed peak. However, its exact state then (even if well known) would reflect prior random climate variability. We thus run 100 years of stochastic forcing *prior* to the 150 year experimental interval. This allows ensemble members to decorrelate from the model's initial steady-state configuration. Since we are only focused on industrial-era retreats, we discard ensemble members whose termini retreat behind the bed peak in the 100-year decorrelation period.

From each aleatory ensemble, we define the probability of sustained retreat as the number of simulations exceeding the 4 km retreat threshold, divided by the total number of simulations (excluding those that retreat during the decorrelation period from consideration entirely). We assess the role of anthropogenic forcing by comparing the probability of retreat between an ensemble with constant mean climate and another ensemble with an anthropogenic trend in the mean climate added to all members (Section 4).

The aleatory ensembles are the key tools of the attribution framework we propose, and are necessary because of the fundamentally chaotic nature of climate variability. A few previous studies have analyzed aleatory ensembles for ice-sheet dynamics, focusing on uncertainty in future response (Tsai et al., 2017; Robel et al., 2019; Tsai et al., 2020), but the present study is the first application to the question of attribution. However, a different category of ensemble methods also exists for quantifying "epistemic" uncertainty in the design or parameter choices within the model. In principle, such epsitemic uncertainty can

be reduced by adding observational or theoretical constraints, whereas aleatoric uncertainty is irreducible due to the lack of predictability of climate variability. These ensemble methods have recently seen wider use for ice-sheet models. Examples include intercomparison projects targeting structural differences between models (e.g., ISMIP6; Nowicki et al., 2016), perturbed-parameter ensembles (e.g., Applegate et al., 2012; Aschwanden et al., 2019; Nias et al., 2019; DeConto et al., 2021), and ensembles of synthetic subglacial topography (MacKie et al., 2021). We also employ ensemble methods in this broader

epistemic category. In the next section, we assess the sensitivity of the probability of retreat to several key parameters by running aleatory ensembles over a range of parameter perturbations (note that this means two layers of ensemble design over two different types of uncertainty). Finally, we also consider an ensemble of different bed topographies (Section 5) in order to investigate attribution in the context of a regional population of heterogeneous glaciers.





We note that the synthetic geometries and climate parameters we use are closer in scale to marine-terminating glaciers in
Greenland than Antarctica. However, though our examples and discussion are mainly oriented towards Greenland's glaciers,
the principles of the ensemble attribution framework constitute a general approach for assessing glacier variability and retreats
near topographic thresholds. These methods could thus be applied to other marine-terminating glaciers in a wide range of
contexts.

## 3   The null hypothesis: noise-triggered retreat

The basic mechanism of glacier retreat on retrograde slopes is well established, and previous modeling studies have investigated
controls and thresholds for retreat (e.g., Nick et al., 2009; Enderlin et al., 2013; Parizek et al., 2013; Catania et al., 2018), as
well as factors that enhance stability (e.g., Gudmundsson et al., 2012; Pegler, 2018; Gomez et al., 2015; Robel et al., 2021).
However, few studies have systematically investigated sustained retreats driven by stochastic climate variability (with some
exceptions, see Mulder et al., 2018; Robel et al., 2019). We thus begin by investigating controls on noise-driven retreat before
turning to the question of anthropogenic forcing.

In the single simulation shown in Fig. 2, stable fluctuations extend right up to the bed peak before sustained retreat. Despite
the instability associated with retrograde slopes, glacier termini near relatively steep bed peaks can be very stable to perturba-
tions (Robel et al., 2021). Ultimately, for attribution, we need to know how stable glaciers are with respect to natural variability,
and we can use aleatory ensembles to assess this. We first consider two different 1000-member ensemble experiments, both
with a bed-peak height of 94 m. The first ensemble is run with frontal ablation variability of $\sigma_{FA} = 15$ m/yr, and contains no
ensemble members that produce sustained retreat in the 150 year period (though note again that minor excursions behind the
bed peak are possible; Fig. 3a). This is consistent with the long wait time for retreat in Fig. 2. However, an ensemble with
greater frontal ablation variability ($\sigma_{FA} = 27$ m/yr) produces a significant number of sustained retreats (Fig. 3b), with a retreat
probability of approximately 15% in 150 years. Note that the ensemble size is reduced in Fig. 3b because more members
produce retreat during the decorrelation period. These two cases illustrate how large ensembles can be used to estimate the
probability of retreat under internal climate variability, but also that the estimated probability is sensitive to model assumptions,
including the statistics of the climate variability.

### 3.1   Which model parameters affect the probability of rapid retreat?

With multiple uncertain parameters that must be prescribed in any model, it is important to consider how sensitive retreat
probabilities estimated from an ensemble will be to such uncertainties. We assess this by running groups of aleatory ensembles,
where each ensemble has slightly different model parameter values (model parameters are still identical between the 1000
simulations within a single aleatory ensemble). We perturb the bed peak height (from 88 m to 100 m), the mean interior SMB
(64 cm/yr to 71 cm/yr), and the friction coefficient (reductions of 2 to 22% from the default value). We vary only one parameter
at a time, but for each group of perturbations, we also apply three levels of frontal melt variability ($\sigma_{FA}$ of 15, 21, and 27 m/yr).



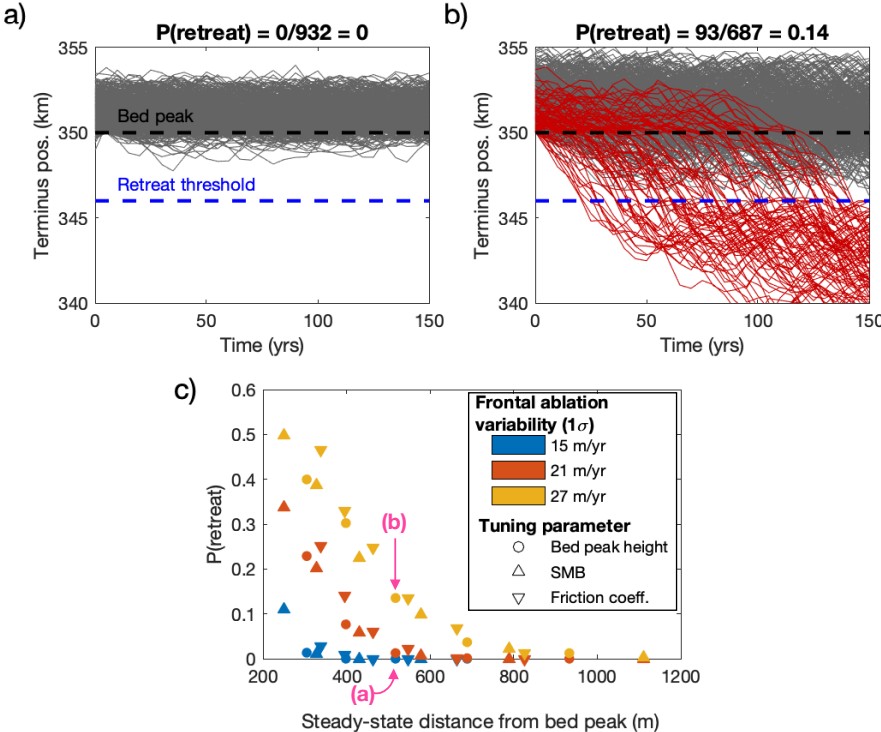

**Figure 3.** Probability of sustained retreat driven by stochastic variability. (a) An aleatory ensemble experiment with frontal ablation variability of $\sigma_{FA} = 15$ m/yr. No simulations produce sustained retreat behind the location of the bed peak (black). Sustained retreat is defined here as retreating past the deepest part of the trough at 346 km (blue). (b) An ensemble subject to stronger variability ($\sigma_{FA} = 27$ m/yr), where approximately 15% of simulations produce sustained retreat (red). (c) Retreat probabilities for a range of bed-peak heights (circles), SMB (triangles), and friction coefficients (downward triangles). Symbols are colored by the range in $\sigma_{FA}$. Arrows indicate the ensembles shown in (a) and (b).

Figure 3c shows the retreat probabilities of 51 ensembles, each with 1000 members, across this parameter space. Retreat probabilities are plotted against the distance between the initial steady-state terminus position (before being forced by climate variability) and the bed peak. While the steady state terminus position is not strictly the same as the time-mean of fluctuations (due to nonlinearities in ice flow; Robel et al., 2018), it still provides a useful metric for how close the system is to the retreat threshold associated with bed topography. From Fig 3c, we can see that the probability of rapid retreat varies primarily along two dimensions; the proximity of the glacier terminus to the bed peak, and the magnitude of noisy forcing. Regardless of which
model parameter is changed (besides the amplitude of forcing), the probability of retreat ultimately collapses onto curves (i.e. the different colored markers in Fig. 3c) that are solely a function of the steady-state distance of the terminus from the bed peak. Similarly, the effect of increasing the climate variability is fairly consistent for a given initial steady state, regardless of which glaciological parameters are perturbed to achieve that state.



Setting up attribution experiments for real glaciers will require prescribing multiple uncertain parameters, and the probability of retreat can change quickly across a relatively small range (Fig. 3c). It will thus be important to evaluate how sensitive estimates of retreat probability are to these uncertainties. While there are many parameters to consider, Fig. 3c shows that the priority should be for sensitivity tests to adequately sample the plausible range of mean terminus position prior to retreat, and of the magnitude of forcing variability. Both of these key constraints will likely require better observations of glacier state and
climate from before the satellite era.

## 3.2    What causes individual glacier retreats?

Beyond providing a means to estimate the probability of glacier retreat, aleatory ensembles also provide large synthetic datasets that can offer further insight into the process of sustained retreat in the context of climate variability. In Fig. 4 we analyze a 1000-member ensemble, run for 500 years with frontal ablation variability that has $\sigma_{FA} = 18$ m/yr. We use the standard
idealized geometry with a bed peak of 90 m. From this ensemble, we find 375 simulations that retreat more than 4 km, which we will refer to as the "retreat group". These retreats begin at random times within each simulation. In order to analyze them together, in Fig. 4a we synchronize the time series such that the year 0 corresponds to the last time that the terminus is forward of the bed peak in Fig 4a. We conduct the same synchronization for each ensemble member's corresponding timeseries of frontal ablation forcing, so that the anomalies causing retreat are lined up (Fig. 4c). Then, for comparison to the retreat group,
we select the simulations that temporarily retreat at least 1 km behind the peak, but then recover instead of retreating entirely. These are also synchronized to the onset of retreat (Fig. 4b, d). Comparing the ensemble-mean forcings for both synchronized groups demonstrates that the frontal ablation anomalies that trigger sustained retreat are not systematically larger than those that allow recovery (Fig. 4e). However, they are on average more persistent into the decades following the onset of retreat. Note that in some individual cases visible in panel (c), this persistence is not a continuous excursion, but multiple positive anomalies
in quick succession.

    The importance of persistence in climate anomalies for triggering sustained glacier retreats is a consequence of the multi-decadal response time of ice dynamics near the terminus (Robel et al., 2018). Since glaciers integrate climate forcing on this timescale, persistent forcing anomalies are more likely to induce retreat that is unrecoverable. Following an initial retreat from the bed peak due to anomalous frontal ablation, discharge fluxes increase when the terminus retreats into deeper water, and
dynamic thinning propagate inland. Such changes are reversible if the forcing anomaly recovers before significant changes in ice flow occur (Fig. 4b). However, the longer the terminus persists in deeper water, the more ice is discharged. At some point, flux imbalances grow to a point where the terminus cannot recover even when the frontal ablation anomaly eventually ends, and thus the marine-ice-sheet instability takes over in driving the retreat.

    The groups of simulations in Fig. 4 reflect only one aleatory ensemble, where the forcing variability was drawn from the same
statistical distribution and had no underlying trend. The variations in the persistence of frontal ablation excursions occur purely by chance, as would be expected in natural, internal climate variability. However, an external forcing trend systematically adds persistence to any timeseries (Fig. 4f). We can anticipate that such extra persistence will affect the probability of rapid glacier retreats, as we explore in the next section.





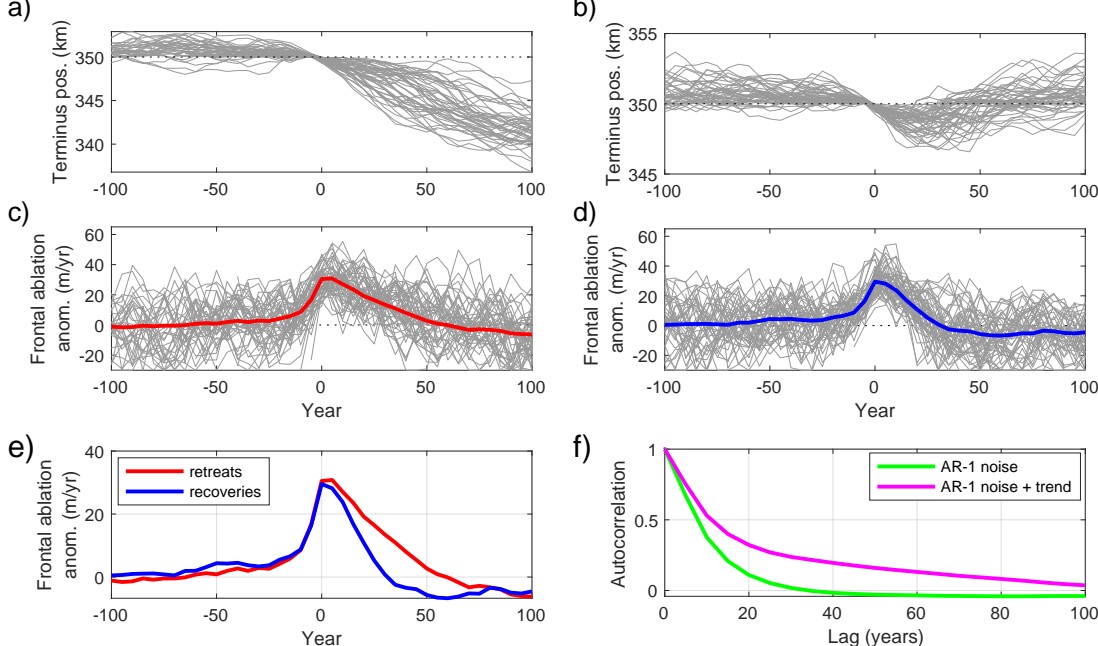

**Figure 4.** Unstable retreats are driven by persistent anomalies in climate forcing. a) Unstable retreats from a 1000-member ensemble, synchronized to the time that retreat begins (only 50 members shown for clarity). b) Synchronized members from the same ensemble that retreated >1 km from the bed peak, but then recovered. c) Synchronized frontal ablation anomalies for the retreat group in (a), along with the group mean (red). d) As for (c), but for the recovery group. e) Group-mean frontal ablation anomalies plotted together. On average, the key difference between unstable retreats and recoveries is the persistence of the anomalies, rather than the maximum at year 0. f) Autocorrelation function of the AR-1 noise used to simulate frontal ablation anomalies (green), and autocorrelation of the same AR-1 noise plus a 150-year trend of similar magnitude to the variability. A trend will enhance persistence on the multidecadal time frames that set apart the retreat and recovery groups in (e).

## 4   Anthropogenic trends and the probability of glacier retreat

Attribution of observed glacier changes depends on comparing simulations with and without anthropogenic forcing. For individual glaciers, this requires an estimate of the local signature of anthropogenic climate change, which is an attribution challenge itself (e.g., Hegerl et al., 2006). The rate and the time of onset of anthropogenic forcing trends are especially uncertain in the polar regions, due in part to a scarcity of long-term instrumental records. We return to this issue in the discussion, but proceed here with a synthetic attribution experiment consisting of two possible anthropogenic forcing scenarios (an early onset

and late onset), and the null scenario of no anthropogenic forcing (Fig. 5a). While these are idealized simulations, multiple forcing scenarios give us a way to assess how the timing of anthropogenic climate changes affect attribution for glaciers.

We consider three hypothetical scenarios of ocean forcing over the industrial era (which we take to be 1870–2020 CE): (1) an increase in frontal ablation ($\Delta_{FA}$) of 24 m/yr applied as a linear trend over the 150 years; (2) the same increase over



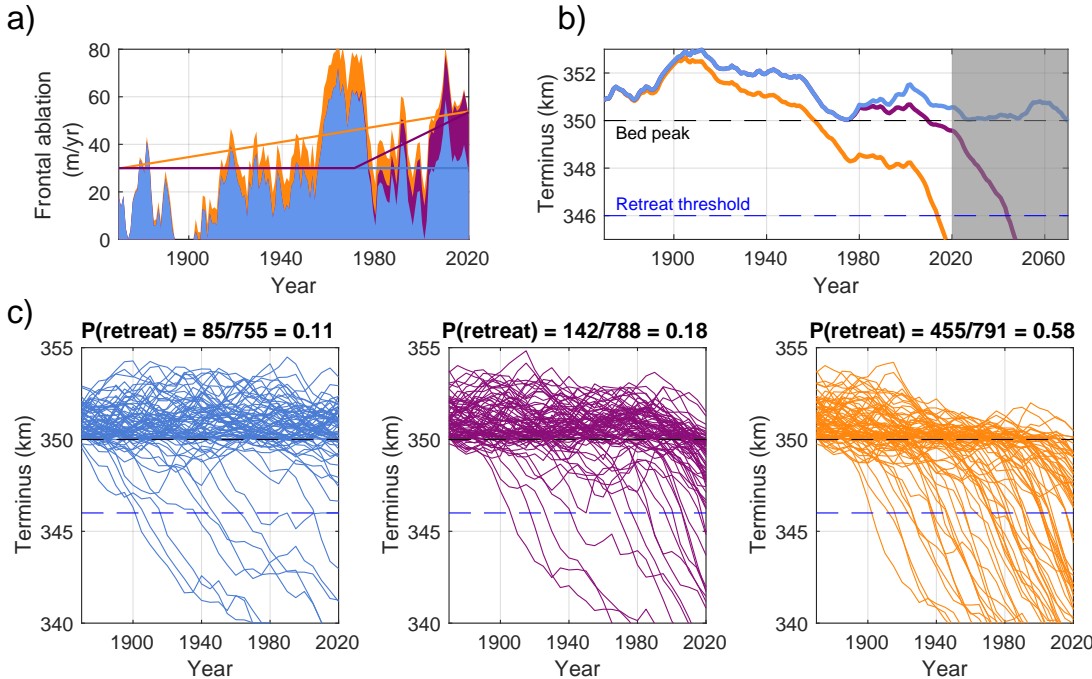

**Figure 5.** The effect of a persistent trend on retreat probability. a) Three forcing scenarios: no trend (blue); an 80% increase in frontal melt over 150 years (orange), and a delayed trend of the same total magnitude, but over 50 years (purple). Solid lines show the forced trends in frontal ablation, while shaded regions illustrate one realization of stochastic variability added to each scenario. b) Glacier-terminus responses to the forcing shown in (a). The plot extends past the 150-year experimental interval to show the long-term glacier response to prior forcing (grey shaded); no additional trend is applied after 150 years. c) Ensemble results (i.e., independent realizations of variability) for these three trend scenarios. For clarity, only 10% of ensemble members are plotted.

only the last 50 years; and the null scenario of no trend (Fig. 5a). Stochastic anomalies are also added to each scenario, with

$\sigma_{FA} = 18$ m/yr and a persistence timescale of 10 years. While not explicitly emulating a particular climate record, this forcing is meant to generally capture the status of climate variables thought to affect ice-ocean interactions. That is, century-scale trends exist (e.g., in ocean temperatures or circulation anomalies), but (multi)decadal variability is comparable in magnitude (e.g., Straneo et al., 2013; Holland et al., 2019; O'Connor et al., 2021). The values here will give an average signal-to-noise ratio ($\Delta_{FA}/\sigma_{FA}$) of about 1.3, which for a single timeseries fails to meet significance at 95% under a one-tailed $t$-test (assuming

decadal persistence). However, note that the particular sequence of anomalies in Fig. 5a has a greater apparent trend, purely by chance.

    Forcing the synthetic glacier (here with a bed peak of $90$ m in height) with the particular frontal ablation anomalies in Fig. 5a, we can see that only the scenarios with an anthropogenic trend drive retreat past the bed peak, though it occurs later for the late-onset trend (Fig. 5b). However, Fig. 5b reflects only one realization of variability, which in this case enhances the overall



trend, muddying interpretation of the retreat. This further motivates ensemble analysis for gaining a more robust understanding for how anthropogenic trends affect the probability of retreat.

For each of the three scenarios, we run 1000-member aleatory ensembles (Fig. 5b). We initialize the experiments and count retreats as described previously. With no anthropogenic trend (left), the null probability of retreat is 11%. In the late-onset scenario (middle), where the anthropogenic trend is concentrated over the last 50 years, the probability of retreat rises to 18%, indicating a modest anthropogenic effect. However, for the early-onset scenario, the probability jumps to 58%—a strong anthropogenic effect.

Why is there such a difference the early-onset and late-onset cases? The integrated and lagged response of ice dynamics again plays a role. The total anomalous melt integrated over the experimental interval is much greater if the trend begins early, and the full dynamic response of a glacier simply has more time to react to an early-onset trend. This is essentially the same principle that differentiates irreversible retreats from reversible retreats in the absence of a background trend (Fig. 4). A positive background trend effectively makes the positive anomalies more persistent, and thus more likely to trigger sustained retreat within the experimental interval. This effect operates over a longer time for early-onset trends. We also note the frontal ablation trend is sufficient to eventually drive retreat from the peak even in the absence of variability (not shown), although in such a case it does not reach the 4 km retreat threshold within 150 years. While sustained retreat is committed even for simulations that do not count as rapid retreats, attribution focuses on the probability of having already observed rapid retreat.

Figure 5b further illustrates these effects. The early-onset trend helps push the glacier beyond the point of recovery when a strong frontal ablation anomalies occur in the middle of the simulation (Fig. 5a). The simulations extend 50 years beyond the experimental interval (shaded), with no further trend after 2020. This shows that the late-onset trend does indeed commit the glacier to a sustained retreat, but it begins during a later phase of positive random anomalies. This is just one realization of variability, but illustrates how persistent trends set up the glacier for retreat and, equivalently, may prevent recoveries.

The key takeaway is that a long-term anthropogenic trend, even if it is weak compared to the noise, can significantly increase the probability of unstable retreats within a given time frame. A higher probability of retreat is expected for trends with a longer duration, or larger magnitude (see supplement for additional forcing scenarios). The attribution of observed retreats can be framed around such increases in probability. However, a statistically weak forcing trend is also an uncertain trend, so defining the counterfactual "no forcing" scenario based on observations is a key challenge here. Figure 5 demonstrates that the onset of forcing is an important consideration, due to the typical response times of ice dynamics. Multiple ensembles could be run to provide bounds on attribution that are consistent with uncertainties in the anthropogenic forcing. We return to this issue in the discussion section.

## 5   Attribution for a population of glaciers

So far, we have focused on how to frame attribution for the observed retreat of a single glacier. However, especially in Greenland, groups of glaciers have retreated concurrently within regions and across the entire ice sheet (Murray et al., 2015; King et al., 2018), although with heterogeneity in retreat duration and extent between individual glaciers (Catania et al., 2018).



Moreover, total Greenland Ice Sheet mass loss depends on the behavior of many marine-terminating glaciers, which would need to be considered for attributing ice-dynamic contributions to sea-level rise. We therefore consider how attribution might
be framed for a population of glaciers. While there are several mechanisms that may cause heterogeneous glacier responses (e.g. local geometry, catchment size, fjord hydrography, basal hydrology) we focus here solely on variations in bed topography in order to demonstrate the attribution framework. However, similar experiments could be conducted in a more complex ice-sheet model including these other factors.

At regional scales, local topography can explain much of the variation in the magnitudes of terminus retreats (Catania et al.,
2018). Similarly, we expect that the pre-retreat distance between glacier termini and bed peaks would vary among individual glaciers, and with it the likelihood of rapid retreat (as shown in Fig. 3). Such glacier-to-glacier variation in the thresholds for rapid retreat may make attribution for glacier populations more statistically robust, since it could be framed in terms of the fraction of glaciers within a region that retreat, rather than the probability of retreat for a single glacier, which may be strongly sensitive to coincidental alignments of random variability and bed topography.

Subglacial topography is known to have self-similar properties over a wide range of length scales (e.g., Jordan et al., 2017). To emulate these characteristics, we use a MATLAB function for generating random surfaces with prescribed self-similarity characteristics (Kanafi, 2021). We consider an initial population of 200 glaciers with random bed topographies. For each glacier, we start with a gentle prograde bed slope ($7 \times 10^{-4}$) and then superimpose topographic variations, unique to each glacier, beyond $x = 200$ km. Topography is generated with a fractal dimension of 0.5 and root-mean-square variation of 80
m, which are realistic statistics describing variations of bed topography in Greenland (Jordan et al., 2017), although we note that the assumption of random variations may fail to capture non-random aspects of topography unique to terminus regions. To avoid aliasing as the glacier model's numerical grid varies in time, we smooth the bed profiles with a 30 m running-mean filter, and also refine the numerical grid in the region of topographic variability to approximately 1 km (the model's grid resolution remains $< 0.1$ km for $\sim 6$ km upstream of the terminus). This smoothing enhances numerical stability of the model,
but we expect it has little impact on the locations of actual glacier stability since it is much finer than the ice thickness (e.g., Gudmundsson, 2003).

To generate a self-consistent glacier geometry on each synthetic topography, we start with a terminus position beyond the random topographic variations (which aids initial numerical convergence), and then reduce SMB so that the terminus retreats toward a new equilibrium within the variable topography. We allow 5000 years for this adjustment, imposing frontal ablation
variability throughout (identical for all glaciers). Some glaciers evolve quickly to new stable positions (e.g. Glacier 21; Fig. 6), while others exhibit multiple punctuated retreats over several millennia (e.g. Glacier 28; Fig. 6), depending on their individual bed topographies. The end result is a population of glaciers with a spread of terminus positions grounded on a variety of prograde slopes and bed peaks (Fig. 6b). We discard glaciers that have less than 10 km of variable topography inland of the terminus at $t = 5000$ yr (light grey in Fig. 6a), leaving a population of 177 glaciers (dark grey in Fig. 6a). While this may bias
the population of considered glaciers towards stability, the number of excluded glaciers is relatively small ($n = 23$).

We then use the glacier states at $t = 5000$ years as the initial conditions for a synthetic attribution experiment. Of these, there is a wide distribution in the distance between the terminus and the nearest local topographic peak (Fig. 6c). The majority are



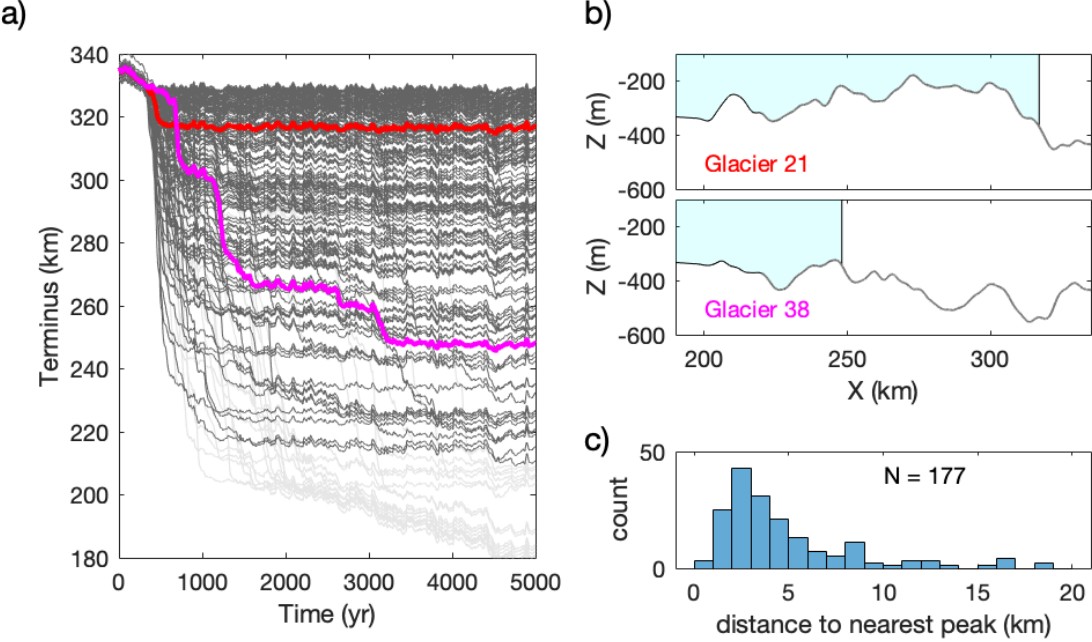

**Figure 6.** Simulating a population of glaciers with randomly generated bed topographies. a) Transient spinup to find quasi-stable initial conditions in random topographies. Two glaciers are highlighted (red and magenta) to illustrate the range of different retreat histories. Light-grey lines indicate glaciers excluded from subsequent analyses. b) Topographic profiles and terminus position at the end of the spinup, for the two glaciers highlighted in (a). Note the very different topography inland of the terminus. Glacier ice is colored blue. c) Distribution of 177 terminus positions at $t = 5000$ years with respect to the nearest bed peak.

within 5 km of a peak, however peaks are defined simply as local maxima of the smoothed self-affine topography. As a result, peaks range from a few 10s to several 100s of meters in height (compared to their nearest troughs), and thus may provide

different degrees of stability. This spinup should not be interpreted as a reconstruction of glacier behavior over the last several millennia; it is simply a procedure to yield a population of quasi-steady termini on random topographies.

We now assess the role of an external forcing trend over a 150 year interval in driving retreat within this population of glaciers. Whereas the transition from stable to unstable terminus positions was simplified by the single bed peak in previous experiments (Fig. 1), transitions can be less obvious with topographic complexity at smaller scales, especially when a glacier

is fluctuating in response to climate variability. Rather than focusing strictly on dynamical stability, which can be very difficult to assess from observations (Robel et al., 2021), we simply set a threshold of 2 km net retreat over the experiment to consider a glacier as "retreated". This choice is arbitrary, but our purpose is to have a common retreat threshold for the population, so that the number of rapid retreats can be compared between forcing scenarios. We show qualitatively similar results with different threshold values in the Supplement.





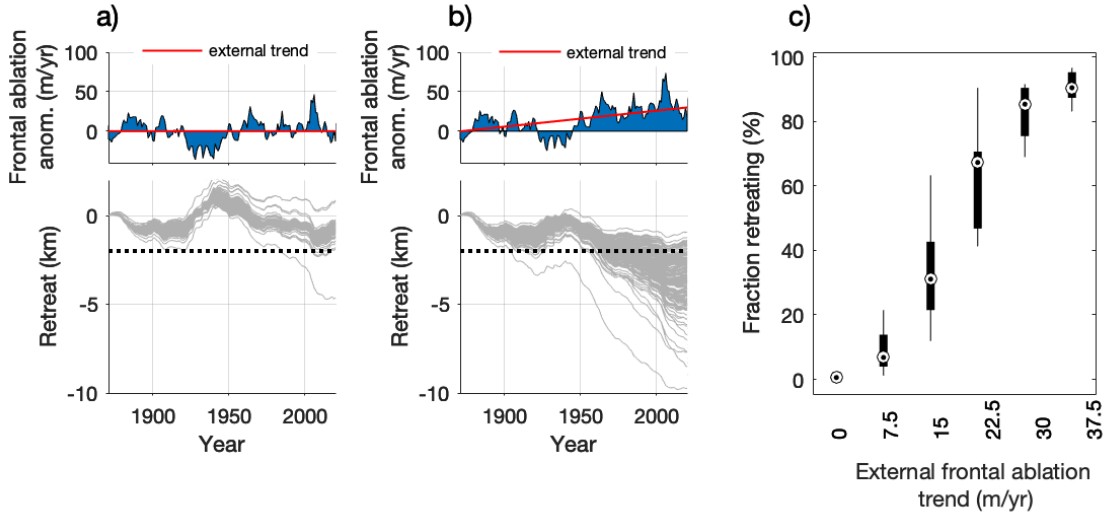

**Figure 7.** Synthetic attribution experiment for a population of glaciers. (a) Terminus change (relative to year 0) for each glacier, all forced with the same frontal ablation anomalies ($\sigma_{FA} = 15$ m/yr, top), which have no external forcing trend. Dashed line shows a retreat threshold of 2 km. (b) as for (a), except with an external forcing trend in frontal ablation, increasing 30 m/yr over 150 years. (c) Box plot showing proportion of glaciers in the population exceeding the retreat threshold, across six external forcing scenarios. "External frontal ablation" refers to the total linear trend over 150 years. For each level of forcing, 20 simulations of the glacier population are run with different realizations of climate variability (i.e., a small aleatory ensemble), with the distribution of retreats indicated with box-and-whisker markers.

We force each glacier in the population with the same frontal ablation anomalies in order to mimic regionally coherent climate variability. With stochastic variability and no external trend, termini fluctuate coherently, though by slightly different amounts depending on how different bed geometries modulate terminus fluctuations (Fig. 7a). However, only one glacier shows a net retreat of $> 2$ km. In contrast, with a frontal ablation trend of 30 m/yr over 150 years (and the same stochastic anomalies), there is an overall retreat trend for the population, with the majority retreating >2 km over the latter part of the simulation (Fig.

7b).

The simulations shown in Fig. 7a and 7b show only one realization of random climate anomalies. This means that the number of large retreats (in both scenarios) depends partly on the particular sequence of random anomalies. To illustrate this approach more generally, we run simulations for the same population of 177 glaciers with six levels of anthropogenic trend, using small aleatory ensembles of 20 realizations of variability for each scenario (Fig. 7c). As expected, the number of rapid retreats in the

population increases with the magnitude of the frontal ablation trend, but there is a spread depending on whether the stochastic variability enhances or reduces the background trend in the latter part of the simulation. This spread is widest for mid-range forcing scenarios, where glaciers are on average closer to topographic thresholds, but where large retreats for many glaciers are still contingent on a contribution from natural variability. With a large enough forcing, however, retreats larger than a given threshold are ubiquitous regardless of the sequence of random variability (Fig. 7c).



Earlier, we showed how a long-term trend primes a single glacier for rapid retreat in the presence of climate variability (Fig. 5). Here, glaciers in the population have different sensitivities and thresholds due to their unique bed topographies. While an external forcing trend generally increases the number of large retreats, modest forcing may first cause a heterogeneous response by pushing some (but not all) glaciers into a range where random variability may trigger rapid retreat. Some glaciers are simply closer to topographic conditions permitting retreat (e.g., as compared in Fig. 6b). However, the failure of some glaciers to retreat

significantly does not necessarily preclude a regional attribution statement. Provided that simulations of many glaciers can be undertaken, attribution could be cast in terms of an increased number of large retreats (or an aggregate metric such as volume loss) compared to the case with no external forcing. Thus, observations of a region in which a substantial fraction of the glaciers retreated would likely be a very strong, statistically robust indicator of an anthropogenic influence, since it is nearly impossible for this to happen in the absence of a trend (Fig. 7c).

## 6   Discussion

### 6.1   Uncertainty in the climate forcing

Our results from large aleatory ensembles show how anthropogenic trends in climate forcing affect the probability of rapid marine-terminating glacier retreat in the presence of bed variations, even when the trend is weak compared to natural climate variability (Fig. 5). The comparison between early- and late-onset trends highlights that a glacier's long-term integration of climate forcing is fundamental to the increased probability of retreat. The early stages of anthropogenic forcing may thus

play an important role in an overall attribution assessment. These effects are very clear in the synthetic experiments, where the difference between ensembles forced with and without trends is simply imposed. However, the integrated effects of early forcing also mean that uncertainty in the onset of trends translates to uncertainty in attribution. When targeting real glaciers, it will thus be important to evaluate assumptions about the onset of trends built into the model simulations.

What do observations tell us about the onset of climate trends in polar regions? Instrumental records spanning the industrial era are scarce or nonexistent in the vicinity of many glaciers, but a few air-temperature records in Greenland extend over two centuries (e.g., Vinther et al., 2006) and show warming trends since the early 19th century. At a regional scale, the multi-proxy reconstructions of Abram et al. (2016) also show an early (∼1830) onset of surface warming in the Arctic, but not the Antarctic. The few long-term ocean records off of Greenland show near-surface (0–40 m) temperature variations roughly

correlated with air temperatures (Ribergaard and Buch, 2008), though strong variability obscures the onset of local trends. Additionally, mid-latitude Atlantic waters, a key source of thermal forcing for Greenland's glaciers (e.g., Straneo et al., 2011), have been accumulating heat since the late 19th century (Roemmich et al., 2012). These records do not alone partition natural vs. anthropogenic effects, but do provide evidence of early-onset trends.

    Comparisons of global climate model simulations with and without anthropogenic greenhouse gas emissions show a signifi-

cant anthropogenic warming signal beginning in the late 19th century, and accelerating in the late 20th Century (e.g., Haustein et al., 2017; IPCC, 2021). However, the timing of global-mean temperature changes may not translate directly to the local changes relevant to glaciers. Melt thresholds can add nonlinearity to forcing trends, changing the time of emergence (e.g.





Trusel et al., 2018). Transient warming is also slower for ocean variables due to the thermal inertia of the ocean, and localized radiative feedbacks produce further regional variations in surface warming, especially at high latitudes (e.g., Armour et al.,

2013). Additionally, the spatio-temporal signature of some forcing agents is relevant at high latitudes. For example, anthropogenic aerosols had an outsized effect in the Arctic and North Atlantic in the 20th Century, and several studies have concluded that they effectively cancelled the greenhouse-gas forcing in the mid 20th Century in the Arctic (e.g., Fyfe et al., 2013; England et al., 2021). This would imply a more step-wise anthropogenic forcing trend, which would be important to account for in attribution experiments. Note that time-variations in aerosol forcing also complicate interpretations of Atlantic multidecadal

variability (Mann et al., 2020). For glaciers sensitive to this variability, these effects would be important to consider when defining the timescales of stochastic forcing in model experiments.

Despite these challenges, it is still possible to get a sense of the external forcing component using output from GCM ensembles. Single-model large ensembles (in which ensemble members differ only due to internal variability) can be used to estimate externally-forced trends at any model grid point (e.g., Deser et al., 2012), and unforced simulations are available for compari-

son. Statistical methods for estimating the forced and internal components in model output and observations also continue to improve (e.g., Mckinnon et al.; Wills et al., 2020)

Our results suggest that assessing uncertainty in the evolution of local climate forcing will be very important for understanding the robustness of attribution statements, because of the way glacier dynamics integrate progressive forcing. Ultimately, attribution statements are always made with reference to a world without anthropogenic forcing, which only exists in models.

In order for an attribution analysis to provide useful insight, the assumptions about this counterfactual world have to be clearly presented, and their limits recognized.

## 6.2 Initial glacier conditions

Uncertainty in the preindustrial glacier state is also important to consider in attribution studies. We have shown that a glacier's initial proximity to a topographic threshold is a key factor in its probability of retreat (Fig. 3), but few terminus records are

available before the satellite era (e.g., Goliber et al., 2021). Moreover, the terminus will have fluctuated due to variability in the preindustrial climate (e.g., Fig. 2), so even if early terminus positions are known, discrete observations still leave some uncertainty in the mean state.

It is worth considering limiting cases and what they would imply for attribution. One limit is a mean preindustrial terminus position far enough from a topographic peak that it is virtually impossible for natural terminus variability to breach the thresh-

old. In this case, an observed retreat from the peak could likely be strongly attributed to external forcing. Note that this would also require a large forced response to drive the terminus toward the peak prior to rapid retreat. The other limit is a terminus position perched right at the peak, so that natural variability may quickly drive retreat. We expect it is unlikely to encounter this limit because such a glacier would have already retreated due to past variability in a stationary climate. The likely targets for attribution studies—that is, those glaciers that have retreated from bed peaks during the observational era, but where the

anthropogenic role is ambiguous—will fall somewhere between these limits. It will be necessary to test the sensitivity of conclusions in an attribution study to the range of preindustrial conditions compatible with available observations. Fortunately,





assumptions about the mean state affect the probability of retreat in the same direction for ensembles with and without trends in forcing (unlike assumptions about the forcing, discussed above). That is, initial conditions closer to a bed peak make retreat more likely whether there is a trend in the forcing or not, but the *difference* between the two scenarios is still a way to assess the
anthropogenic role despite uncertainties in initial conditions. This would also apply to other parameters common to the scenarios with and without trends, such as the magnitude of stochastic climate variability. Nevertheless, improved constraints on pre- or early-industrial era glacier state remain an important goal for providing context for recent changes. To extend observations of glacier state before satellite or aerial campaigns, trimlines, marine sediments, and bathymetry may offer useful constraints (Andresen et al., 2012; Csatho et al., 2008; Smith et al., 2017).

Finally, it is also worth considering what an ice sheet's long memory of Holocene climate variations implies for the stability of preindustrial termini with respect to bed peaks. An important and broad question remains as to why glacier termini should be near topographic thresholds to begin with. On one hand, it may seem unlikely to find any glaciers near thresholds if there is a non-trivial probability that random variability will eventually drive retreat. On the other hand, glacier termini may be transiently persistent at bed peaks even without long term stability (Robel et al., 2021). As an ice sheet responds to natural
climate variations (at all timescales) and its margin evolves over variable topography, one might expect to find most termini near peaks at any given time. Indeed, we observe a similar condition in our model spinup with synthetic random topographies (Fig. 6), although this was not strictly intended to emulate past climate. It is also important to consider here that glaciers affect their landscape via erosion, sedimentation, and lithostatic deformation, and so topography may not be random with respect to the terminus. For example, proglacial sedimentation or the character of over-deepened topography immediately upstream of
the terminus might yield clues as to how persistent termini have been, which could help bound the probability of noise-driven retreat. Investigating these geomorphic processes in the context of climate variability could help refine our view of terminus stability both in the preindustrial and modern climate.

### 6.3    Defining glacier "retreat"

Our numerical experiments focus on the general phenomenology of rapid marine-terminating glacier retreats in the context of
climate variability. Accordingly, we assume certain thresholds for retreat in the simulations. However, for analyses of specific glaciers, it will be necessary to define what aspects of the observed change are the target for attribution, and therefore what counts as reproducing the glacier retreat "event" in simulations. For probabilistic attribution, it is generally the case that the more narrowly an event is defined, the harder it is to attribute to external forcing because fewer simulations will resemble the observation in detail (van Oldenborgh et al., 2021). For example, if bed topography permits sustained retreat for many decades
once initiated (i.e., lacking stabilizing topographic features), then the cumulative retreat observed so far depends partly on when retreat was triggered.

The varied timing of terminus retreats in Greenland illustrates this issue. Before the widespread retreats of the last few decades, many marine-terminating glaciers also retreated following rapid Arctic warming in the 1920s and 30s (Bjørk et al., 2012; Vermassen et al., 2019; Andresen et al., 2012). Some, such as Upernavik and Kangerdlussuaq glaciers, have retreated
nearly monotonically through deeper bathymetry since then, although with some variation in the rate of retreat (Khan et al.,





2014; Vermassen et al., 2019). For attributing early-onset retreats, one might focus on the full observed change, assessing the number of stochastic simulations that reproduced the full magnitude of observed retreat. This would likely select for stochastic anomalies that trigger retreats early in the simulation, reducing the relative importance of long-term trends. Alternatively, one could take a binary approach, focusing on any sustained retreat over some threshold initiated in the simulation period.

This would admit later-onset retreats into the analysis, allowing anthropogenic trends in the forced ensemble to have more effect. However, in such a case, the full magnitude of observed retreat would not necessarily be part of the formal attribution assessment, and this would have to be clearly stated. It is reasonable to hypothesize that retreats triggered early in the industrial era were more contingent on natural climate anomalies compared to more recently initiated retreats. Some glaciers in West Greenland retreated abruptly in the early 2000s and have since stabilized on new bed peaks (Catania et al., 2018), offering very

discrete changes that would be potentially easier to define as targets for attribution experiments. Future work might compare these recent retreats against early-onset, more continuous retreats.

These issues illustrate possible tradeoffs associated with defining the target glacier-retreat event. Once again, attribution is inherently contingent on the assumptions and framing of the analysis. Choices on how to define the observed change will affect the final quantitative elements of the attribution statement (e.g., how much more likely was a given retreat made by

anthropogenic changes). Our main point here is that such choices have to be carefully considered and clearly communicated in future attribution studies.

## 7   Conclusions and outlook

In this paper, we have proposed a framework for attributing rapid marine-terminating glacier retreats to anthropogenic forcing trends. These glaciers may exhibit threshold behaviors associated with their bed topography, which adds ambiguity to the

cause of observed retreats because it is possible for natural climate variability alone to trigger sustained retreat from a bed peak (Fig. 2). Thus, we have proposed framing attribution in terms of the probability of retreat under natural variability, comparing scenarios with and without some anthropogenic forcing signal. The probability of retreat can be estimated using large ensembles of glacier model simulations, each with independent realizations of climate variability (Fig. 3). This approach to attribution is different from an "either-or" question of whether retreats reflect variability or trends. Focusing on the likelihood

of retreat embraces the role that internal climate variability appears to play in the timing of observed rapid retreats, but still provides a way to quantify anthropogenic effects.

We conducted synthetic attribution experiments for single glaciers and for a population of glaciers with different geometries. For a single glacier, attribution is based on comparing at least two aleatory ensembles: one (or more) with an anthropogenic forcing trend common to all members, and one with no anthropogenic forcing. For a regional population of glaciers, we would

expect variations in how close preindustrial termini were to topographic thresholds. Using a population of synthetic random topographies, we showed that forcing trends increase the fraction of glaciers in the population that cross topographic thresholds and undergo major retreats (Fig. 7). This could thus be a metric for regional attribution in settings where observed retreats have been heterogeneous.





The results from ensembles are sensitive to topographic boundary conditions, glacier-dynamical parameters, and the statistics of the climate forcing. However, our sensitivity tests suggest that these factors affect probability of retreat along two key dimensions: the glacier's proximity to a topographic threshold, and the magnitude of its natural fluctuations (Fig. 3c). It will be critical to assess the sensitivity of attribution statements to parameter uncertainty, but it may be sufficient to test along the plausible range of these two dimensions rather than conducting an exhaustive parameter sweep across many more dimensions. This may help to devise more efficient sampling strategies for uncertainty quantification studies using models that are more computationally expensive compared to the relatively simple model used here.

Most importantly, these ensemble experiments show that even modest anthropogenic forcing trends could have a large effect on the probability of retreat, especially on century and longer timescales. Our simulations are idealized, but we expect this is a robust result because it reflects the fundamental timescales over which the ice dynamics integrate and respond to climate variability and trends (Robel et al., 2018; Christian et al., 2020). It is important to consider these century-scale responses even when retreats appear correlated with short-term climate fluctuations. The total effect of a long-term climate trend is not limited to making short-term climate anomalies more extreme, but also includes the preceding decades of forcing, which gradually push the terminus closer to the topographic threshold than it otherwise would be. As a result, the probability of retreat in a given period depends strongly on the onset and duration of external forcing (Fig. 5). The takeaway is that there are firm physical grounds for hypothesizing that a century or more of anthropogenic forcing has affected the probability of rapid terminus retreats. A probabilistic framing, enabled by ensemble simulations, is a way to quantify an anthropogenic effect even if individual retreats coincide with natural climate fluctuations.

As discussed in the previous section, uncertainties in a glacier's preindustrial position and in the onset of anthropogenic forcing pose fundamental challenges for attribution. However, despite these gaps, our view is that sufficient mechanistic understanding and observational constraints exist to begin pursuing ensemble-based attribution assessments on well-observed glaciers. A logistical challenge to confront is the computational cost associated with large ensemble simulations. We propose that a hierarchical approach may be a way forward. Many thousands of simulations are feasible for 1-D models as we have used here. These can be used to explore a wide parameter space, including different assumptions about initial and boundary conditions. These insights can then help constrain the use of 2-D models that will capture the key geometric controls. Large aleatory ensemble simulations have recently been run using state-of-the-art models (Robel et al., 2019), so we expect that the feasibility of their application to attribution studies will only continue to improve in coming years. Even if attribution statements have large uncertainty bounds at first, pursuing such studies may still clarify remaining challenges and offer insights into glacier variability, just as the broader pursuit of attribution in climate science has uncovered insights about climate variability and sensitivity. The ability to make quantitative assessments about the role of anthropogenic forcing is an important benchmark in the study of natural systems. Formal assessments would help focus discussion of recent and ongoing cryospheric change, both within the scientific community and in the public.



*Code availability.* Code for the flowline glacier model is available at https://doi.org/10.5281/zenodo.5245271, and additional scripts for the ensemble analysis and figures are available in a public repository at https://github.com/johnerich/XXXX (will be added in final version).

*Author contributions.* All authors contributed to the study design and interpretation of results. JEC conducted the analyses and wrote the manuscript, with input from all authors.

*Competing interests.* The authors declare no competing interests

*Acknowledgements.* JEC was supported by a postdoctoral fellowship from the University of Texas Institute for Geophysics. We acknowledge the Georgia Institute of Technology Partnership for an Advanced Computing Environment for computing resources, and we thank Ziad Rashed and Vincent Verjans for insightful comments and suggestions.



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
