# Peer review of "A probabilistic framework for quantifying the role of anthropogenic climate change in marine-terminating glacier retreats"

_The Cryosphere, 2021_

## Author Comment (AC1)

**Author responses for tc-2021-394, "A probabilistic framework for quantifying the role of anthropogenic climate change in marine-terminating glacier retreats"**

**KEY**
Reviewer comments
Our responses
> *New or **adjusted** text*

**Reviewer #1**

review of "A probabilistic framework for quantifying the role of anthropogenic climate change in marine-terminating glacier retreats" by Christian et al.

This study considers the question of how to quantify the impact on glacier retreat of externally-forced climatic trends, in the context of internal climate variability. The study uses large ensembles of simulations of a simple glacier model to illustrate several points. (The members of each ensemble each having a different realisation of random climatic variability.) One conclusion is that the influence of a climatic trend on a single glacier geometry may be quantified by comparing ensembles of simulations with and without the trend. Another conclusion is that the influence of a climatic trend may be quantified by considering a population of glaciers with different geometries, all under a common forcing. These conclusions may also be stacked, considering a population of glacier geometries under a population of trends. The results illustrate a variety of philosophical and physical considerations that must be addressed in order to make statements about whether contemporary ice sheet retreat is linked to human behaviour.

My assessment is that this is a very nice study that deals elegantly with an important and very challenging problem. The paper is extremely lucid and thought-provoking throughout, and I feel it has taught me a lot.  One could perhaps argue that the conclusions are not particularly surprising, but I think that is a very good thing – in my opinion all the best science leads the reader to an understanding that feels intuitive and ties everything together.  In this paper, the authors have bought some of the key principles of climate-change attribution to glaciology, and I think the glaciology community could benefit hugely from taking these on board.  Further, I feel that the strong nonlinearity of the glacier dynamics studied here means that this attribution work has features that are rarely found elsewhere in the climate change literature.

I offer a few comments below but I have no reservations about this paper.  I think the authors should be allowed to consider my comments below and respond to them however they wish, including ignoring them entirely.  I'm happy to read the responses but don't require this prior to publication.

Many thanks for this thoughtful and encouraging review. There are some great points raised below, and we learned a lot by addressing them - especially regarding the behavior of probability within the ensembles. Please find our responses below:

Comments (in order of appearance in the text)

Introduction: The introduction frequently mentions Antarctica, but after a while in the following methods section it became clear that the simulations here only explicitly deal with marine-terminating glaciers.  The ice streams of main interest in Antarctica all have ice shelves.  I feel the paper could be clearer on the extent to which the results apply to the case in which an ice shelf buttresses the glacier.  Perhaps the authors can state at the very beginning of the paper that they don't study that case, but that the ideas are broadly applicable, and maybe return to that in the discussion.

We agree it's worth noting this early on. We've added the following text to the introduction:

> It is worth noting here that our model simulations do not include floating ice shelves, which can alter the thresholds for dynamic instability via lateral buttressing (e.g., Gudmundsson et al 2012). We focus instead on cases where stability is a function of bed topography, which simplifies our analysis of glacier variability near instabilities. Accordingly, we orient model parameters and the discussion of results around marine-terminating glaciers in Greenland, where floating ice plays a lesser role than in Antarctica. However, inasmuch as strongly buttressed glaciers and ice streams are still subject to climate variability and may be prone to instabilities, many of the fundamental points for attribution could be adapted for such settings (albeit with additional ice-shelf dynamics to capture).

Section 2: I was a little confused about what the authors mean by frontal ablation.  To me, 'ablation' means melting.  However, maybe this is meant to be a combined melting+calving rate? But then, the model does not represent any floating ice and the ice is calved wherever it goes afloat, at a rate equal to the flow speed.  I assume the ablation rate is applied to the face after it has been calved at floatation.  I guess the ice retreats happen because an increase in ablation displaces the terminus back a little, then the ice speeds up, thinning the ice, and causing further retreat of the floatation line. So, I am a bit confused.  I assume I am just showing my inexperience here as this is a well-established model, but perhaps the authors can discuss this a little in the paper as I think the mechanism is relevant.

The frontal ablation is meant to be very general, and agnostic regarding melt or calving (we feel "ablation" is sufficiently general to include multiple mass loss processes). The sequence described above is more or less how we interpret it as well - we've added a bit more description of the process where frontal ablation is introduced in Section 2.1, and note that the main point is to introduce a time-varying ocean forcing.

*We simulate ocean forcing very generally by adding a frontal ablation term to the mass-conservation equation at the grid point closest to the terminus (see Supplement for numerical implementation).* **This frontal ablation term amounts to an additional output flux beyond that dictated by the local ice velocity. Compared to the case without the frontal ablation term, this results in a retracted terminus with a steeper surface slope, such that ice flow balances the additional output flux.** *For real glaciers, flux anomalies could be driven by variable calving or submarine melt, or a combination;* **here we simply interpret these as flux anomalies at the terminus driven by variable ocean conditions. The most salient aspect is that frontal ablation is a localized flux term that we can force to vary in time.**

Section 3: In Figure 3a, why do 68 of the 'natural control' experiments retreat during the decorrelation period that precedes the time period shown in the figure (i.e. only 932 are left out of 1000 started)? Does this imply the period shown, in which no retreats occur, is not representative? I also note that the simulations in panel b are retreating right from the start of the period shown, so presumably these plotted retreats are just a continuation of the population of retreats that occured within that ensemble's decorrelation period? Do these matters imply that the conclusions drawn in this section are dependent upon the decorrelation period chosen?

This comment raises several interesting and important points, and we have run a number of additional simulations to investigate these issues.

First: for the case of fig. 3a in particular, the 68 excluded did not exhibit sustained retreat during the decorrelation period, but had temporary excursions landward of the peak. So this is a consequence of estimating the probability of retreat only from those simulations with termini seaward of the peak at t=0. As we note, this is done to focus only on sustained retreats within the 150-yr interval, but it does have the consequence of eliminating some "stable" fluctuations, which is most evident in the special case of zero sustained retreats. One could set the threshold back from the peak, though in cases with unstable retreat, this would then admit some that started before the experimental interval. The timescales of glacier fluctuations and retreat make it unclear exactly when retreat becomes unstable, so any threshold will be somewhat arbitrary. We thought the clearest would be to base the analyses on the termini seaward of the bed peak at t=0.

As to the broad and important whether the probability varies with time: We ran several long (2000 year) ensemble simulations to better understand this, and made several observations that motivate some changes to the text and figures.

Primarily, we found that a longer initialization period is needed to avoid the effects of noise induced drift. The drift is such that the mean terminus position under variability (estimated from the ensemble-mean of simulations without retreats) is further from the bed peak (seaward) than

the steady-state, which makes retreat slightly more likely following the start steady state (figure below). This effect persists longer than we initially realized, so we have re-run the ensemble simulations for Figs 3 and 5 with a longer initialization period (250 years) and re-calculated the probabilities. The retreat probabilities are slightly lower, but the plots are qualitatively the same and the overall conclusions are not affected.

Even in the absence of noise-induced drift, it is important to understand whether the probability of retreat is constant in time. We are estimating the probability as $n_r / n_s$, where $n_r$ is the number of retreats in the experimental interval and $n_s$ is the number of ensemble members that have survived the initialization period without retreating. That is, $n_s$ diminishes as more and more glaciers retreat. Note that this probability is fundamentally a conditional probability - i.e., the probability of retreat in the experimental interval, conditioned on the fact that a glacier hasn't yet retreated. We'd argue this conditional probability is most relevant for the attribution question focused on some particular observed retreat. However, it is worth contrasting it with the joint probability $n_r / N$, where $N$ is the total ensemble size (i.e., the probability of any of the $N$ ensemble members surviving the experimental interval, *and* then retreating during the observational interval).

In these additional ensemble simulations, we calculate both $n_r / n_s$ and $n_r / N$ over progressive 100-year intervals of a 2000-year, 5000-member ensemble run (figure below). The probability is enhanced in the first couple of centuries, before the noise-induced drift stabilizes. $n_r / n_s$ is fairly stable, but with some caveats. There is some jitter due to sampling error (which we address in a later response), as well as a long-term increase over 2000 years. We infer that the latter is because the variance of glacier fluctuations is under-sampled initially, due to the millennial timescales of interior ice dynamics. The ratio $n_r / n_s$ also becomes less stable as the ensemble size decreases. Note that $n_r / N$ continually decreases; the rate of retreats drops off with the number of surviving ensemble members.

So, there are caveats for short initialization periods (noise-induced drift), and drawbacks for very long initialization periods (the effective ensemble size diminishes, and computational demands are higher). We think a 250-year initialization period is a reasonable tradeoff, even if it doesn't fully account for the longest timescales of ice dynamics. $n_r / n_s$ is fairly stable over this time frame, as both $n_r$ and $n_s$ drop off as more glaciers retreat. (Note that this behavior approximates a Poisson process, where the probability of an event occurring in some fixed interval is constant; of course each glacier can only retreat once, but across the ensemble, the (conditional) probability of retreats per unit time would be constant for a pure Poisson process. There are some caveats due to long glacier memory, but it can still be a useful statistical model to keep in mind.)

We have revised the description of the ensemble methods to clarify discussion on the noise-induced drift and how probabilities are treated. We will also add the figures below to the supplement, to describe the noise-induced drift and the long-range test of retreat probabilities.

*Rather than starting all simulations with a strictly steady-state glacier, we initialize simulations with a 250-year period of stochastic forcing. This is necessary because of noise-induced drift that occurs at the onset of stochastic forcing, due to nonlinearities in ice dynamics (e.g., Robel et al., 2018). Indeed, we find that the steady-state grounding line position is closer to the bed peak than the long-term mean under noisy forcing, slightly enhancing the likelihood of at the beginning of simulations initiated from steady state (supplemental figure SN).*

*From each aleatory ensemble, we estimate the probability of sustained retreat as the number of retreats within the 150-year experimental interval (as defined above), divided by the total number of simulations with termini seaward of the peak at the beginning of the interval. That is, we are fundamentally focusing on a conditional probability of industrial-era retreat (i.e., conditioned on the glacier not having already retreated). We ran several long ensemble simulations to assess how this conditional probability varies in time, and find it to be fairly stable after the noise-induced drift decays, though with some additional caveats at millennial timescales (see supplement and figure SN). We assess the role of anthropogenic forcing by comparing the probability of retreat between an ensemble with constant mean climate and another ensemble with an anthropogenic trend in the mean climate added to all members (Section 4).*

**Noise-induced drift:**

[Figure]

*Figure SN: (a) Noise-induced drift can be illustrated via the ensemble-mean terminus position (blue line) in cases with no unstable retreats (which would bias the mean). The drift decays over roughly century timeframes. (b) as for (a), but with a slightly lower bed peak, which moves the system closer to the threshold. Note that a few retreats occur initially, but none after the noise-induced drift decays. This illustrates how the probability of retreat is inflated shortly after starting from the steady-state initial condition, which is closer to the peak than the long-term mean. The same effect occurs for cases with non-zero long-term probabilities of retreat.*

*Long-run probability test:*

[Figure]

*Figure SN: (a) Probabilities of retreat over 100-year intervals in a 2000-year, 5000-member ensemble run, with a bed peak height of 94 m and $\sigma_{FA}$ = 15 m/yr. Blue markers track the conditional probability (the metric used in our analyses). Red markers track the joint probability, which decays as more and more ensemble members retreat. Note the effects of the initial condition in the first century or two, and the long-term rise in conditional probability (b) As for (a), except with $\sigma_{FA}$ = 21 m/yr. (c) Cumulative retreats (blue) and non-retreated glaciers (red) throughout the 2000-year ensemble simulation with $\sigma_{FA}$ = 15 m/yr. (d) As for (c), except with $\sigma_{FA}$ = 21 m/yr.*

Section 3.1: When varying model parameters, the authors find that the probability of retreat is a function of the distance of the steady-state terminus from a bed peak.  To be clear, are the authors saying that they think this relationship is causal – i.e. a larger displacement of the terminus is required to reach the peak and trigger retreat – or just a correlation – i.e. the underlying physics of the glacier have been changed in such a way as to enhance instability?

Yes - we think the main effect is the proximity to bed peak. Changes in the underlying physics might also play a role in stability, and this probably accounts for some of the spread in the curves in Fig. 3c. But the overall shape of the curves suggests that getting the preindustrial proximity to peaks is the first-order issue. We've added a sentence to clarify:

> *It is possible that parameter perturbations can affect dynamical stability in other ways (e.g., Parizek et al. 2013), but the similarity of these curves for qualitatively different*

*parameters (sliding, mass balance, and bed geometry) indicates that in this case, proximity to the bed peak is the main effect.*

Section 3.2: (line ~260) I didn't quite follow the physics here. I naively feel that an increase in discharge could enhance the advection of thicker ice towards the bedrock high, hence advancing the terminus. Is it always the case that an increase in discharge is the crucial destabilising factor?  I guess it is actually an increase in ice divergence that is needed to thin the ice and retreat the terminus? By the way I am aware of other literature that draws relevant conclusions concerning the frequency response of ice streams to climatic perturbations (e.g. 10.1098/rspa.2012.0180, 10.1002/2017GL075745) though that is only in the ice shelf-buttressed case.

The increase in discharge was meant to refer to that occurring due to retreat into deeper water (once the terminus has crossed into the region of reverse slopes), as opposed to the initial forcing. In that sense the increase in discharge is crucial to the MISI mechanism, but we agree the initial perturbation could be a different type of forcing, e.g. a drop in surface mass balance which would drive retreat by decreasing flux from the interior. We've reworded this paragraph to clarify the sequence:

> *The importance of persistent climate anomalies for triggering sustained glacier retreats is related to the timescales of transient ice dynamics. Consider an initial terminus fluctuation driven by anomalous frontal ablation. If the terminus retreats past the bed peak and into deeper water, discharge will increase due to the strong dependence of ice flux on grounding-line thickness (Schoof 2007). Independent of the initial forcing, this drives dynamic thinning and further retreat (i.e., the marine-ice-sheet instability mechanism begins). These changes are not instantaneous; ice flow near the terminus evolves on multidecadal timescales (Robel et al., 2018), so retreat is reversible if the climate forcing anomaly recovers before significant changes in the inland ice flow occur (Fig. 3b). However, the longer the terminus persists behind the bed peak, the more interior ice is lost to the increased discharge. At some point, dynamic thinning and retreat will proceed to the point where the terminus cannot recover even if the initial forcing reverses, and thus the marine-ice-sheet instability takes over in driving the retreat.*

Section 4:  (line ~302) I didn't quite click with the language that a background trend makes the positive anomalies more persistent.  Which is more important:  the slow thinning of the glacier that I assume accompanies the (quadratic?) time-integrated ablation anomaly, or the fact that any given positive ablation anomaly is larger?

Our intention was to relate that a trend increases the duration of (positive) melt excursions from the preindustrial mean. But we agree the language could be clearer, as this is somewhat different from a purely statistical notion of persistence. Whether the integrated or instantaneous change in anomalies is more important is also great question. Both should play a role during a

trend, but we agree it isn't immediately clear from these experiments which dominates, so we've run some additional analyses (figure below) to isolate the integrated component. It plays a large role, but we agree the direct effect of simply making ablation anomalies more positive should be mentioned too. We've re-framed the paragraph slightly to clarify these two effects, and include the additional figure in the supplement.

*Why is there such a difference the early-onset and late-onset cases? There are two main effects to consider. First, an external forcing trend makes all positive frontal ablation anomalies more extreme. For the late onset trend, there is simply a shorter window in which more-positive anomalies affect the probability of retreat. Second, the response timescales of of ice dynamics also play a strong role. A glacier's response lags forcing on century timescales, so even if the final magnitude of the trends are the same, the earlier-onset trend will push the average terminus position closer to the threshold within the experimental interval. This makes random variability more likely to trigger sustained retreat. We compared these two effects by assessing the probability of retreat only after trends of varying duration, and found that the lagged dynamic response indeed plays a large role (Supplemental figure X). This is essentially the same principle that differentiates irreversible retreats from reversible retreats in the absence of a background trend; the glacier response reflects the forcing anomaly integrated over decades or longer (Fig. 4).*

[Figure]

*Figure. SN: One way we can isolate the effect of a glacier's integrated response to a trend is by assessing the probability of retreat only after different forcing scenarios. Here we focus on the 50 years after four different forcing scenarios: a) No trend; b) a step change at 2020; c) a 50 year trend from 1970-2020; d) a 150-year trend from 1870-2020. The total changes in frontal ablation are the same, so that after 2020 the distribution of frontal ablation is the same in each scenario (with the exception of (a)). Differences in the probability of retreat from 2020-2070 (green boxes) therefore show the effects of past forcing, namely that long-term trends push the*

*average terminus position closer to the bed peak. Note that the effective ensemble size drops off under long term trends because many retreats occur during the trend, but the overall effect of the trend is clear.*

Section 4 General: The paper discusses the difficulty of defining a retreat metric, which I fully sympathise with.  In this section, the paper determines the effect of a climatic trend on the probability of a retreat of a given distance within a fixed time frame – e.g. before the end of a 150 year run.  Under this approach the probability is variable and the time frame is fixed.  I wondered if the authors had also considered the inverse approach – asking what is the 'time-to-emergence' of a fixed probability of retreat.  E.g. if we choose to be interested in a 50% probability of retreat, the authors could determine how long any given trend would take to induce such a probability (compared to a no-trend scenario). This would have the advantage that the outcome is not a function of the arbitrary duration considered (replacing that with the arbitrarily chosen probability).  I recognise that the approach currently taken may be more appropriate to historical attribution, and the time to emergence idea is usually used for projections.  My guiding principle here is that under ANY nonzero climatic trend, eventually ALL glaciers will have retreated.  So, to me, a time-to-emergence metric reflects that situation.

The time-of-emergence metric is an interesting alternative approach, and could potentially offer additional intuition surrounding the general problem of variability near thresholds. The complementarity between probability and wait time is a useful concept to consider, as the retreats bear some resemblance to a Poisson process. And as you note, such an approach might be highly relevant to comparing the anthropogenic vs. stochastic effects on the timing of future retreats, for glaciers now poised on further-inland bed peaks, or those whose current stability is ambiguous. However, we think it is best to keep the focus in this study on the fixed interval-approach, as it focuses specifically on the whole industrial era. Although the exact onset of glacier-relevant forcing is uncertain (as we discuss), the industrial era (e.g., late 19th Century on) does provide a concrete interval to focus on and avoids the need to pick an arbitrary probability threshold.

Section 4 General:  Is there a significance test that needs to be applied here? If we see a difference in retreat probability between ensembles with and without a trend, or with two different trends, surely we need to determine that difference is statistically robust?  I cannot immediately think what would be the appropriate test, but I assume it would tell us what ensemble sizes are needed to establish a given retreat-probability difference between two ensembles at a stated confidence level.

This is a good point - there is certainly some sampling uncertainty in these probability estimates and it is useful to understand how it depends on ensemble size. Briefly, we can treat each ensemble member as a Bernoulli trial where "success" is a retreat during a certain interval. The probability of success depends on the chosen interval and model parameters, but for a given set of choices the probability is identical for each simulation. Estimating the probability from N

simulations (trials) is a common problem, with standard formulae for estimating the error (which is typically proportional to 1/sqrt(N)). For the conditional probability which is our focus, we have to combine the error from the spinup period and the experimental interval. We've arrived at a metric for the standard error, which we will derive in the supplement. For the range of parameters in our simulations, the differences between forced and unforced ensembles are highly significant. For example, in the new simulations for Fig. 5, the unforced scenario has a 0.04 probability of retreat, and we estimate the standard error at 0.007. The effect of the trend is thus far beyond sampling error.

We want to stress that this degree of significance ONLY reflects statistical sampling issues, and no other uncertainties. We expect that uncertainties in model parameters/physics or in the forcing will likely dominate in attribution assessments for real glaciers, and these will need their own treatment. For this reason, we are reluctant to add confidence intervals to the probabilities in the panel titles (e.g., Fig 5) since it would be hard to convey the context that it is only one source of error - we want to avoid giving the impression of a highly precise method to readers just skimming the figures. However we will note these issues in the main text, and provide detail on estimating sampling error in the supplement.

New text in section 5:

> *We estimate the sampling uncertainty for these probability estimates using standard formulae for ensembles of independent trials (see supplement). We find standard errors of roughly 0.01–0.02 for the ensembles in Fig. 4, making the effect of the trend far greater than sampling error. However, we stress that this is only one source of error, and uncertainties in model parameters can have a much larger effect (Fig. 3).*

Section 5 General: This section assumes that all glaciers in the population have identical climatic forcing, which seems a little restrictive to me. For example in Greenland, all glaciers experience similar atmospheric conditions and far-field ocean forcing, but that is quite different to saying they have the same SMB and frontal ablation rates, which are determined by very local features such as ice topography, fjord geometry etc. I believe the logic assumes that if neighbouring glaciers have different retreat history, that can only be caused by terminus bed geometry, which I don't believe is always the case.

We completely agree this is a major simplification. The main goal here is to consider variety in glacier's proximity to topographic thresholds. We can achieve this solely with a set of random topographies, and since we aren't focused on simulating a specific glacier or region, we didn't see much benefit in adding variations in each glacier's climatology as well. We do note at the beginning of the section that there are other factors leading to heterogeneity, but we agree the simplification should be more clearly flagged. We've added some text to emphasize this here:

> ***The synthetic glaciers we present below thus do not represent the full spectrum of glaciers that could be found in a region, though** similar experiments could be conducted in a more complex ice-sheet model including these other factors.*

And also where we state that the forcing is identical for each glacier:

> *We force each glacier in the population with the same frontal ablation anomalies in order to mimic regionally coherent climate variability. **This neglects a number of factors that can cause ocean forcing to vary widely between individual glaciers (e.g., Straneo et al., 2011; Wood et al., 2021) but our focus remains on simplified experiments to illustrate attribution---here with a variety of topographies.***

Section 5 Figure 7: I was initially surprised that the ensemble in panel a has only one member that advances. I believe this is telling me that there is a statistically significant internally-generated trend in the climatic forcing (towards retreat). This means that the ensemble is 'primed' such that when the external trend is added in panel b, lots of glaciers retreat. This is useful for illustrative purposes, but it is not mentioned and I think the authors should be open about this situation. They could potentially add an internally generated trend line to panel a. They could add a red dot to panel c illustrating that this chosen realisation sits above the mean fraction retreating for 30 m/y (I assume). Probably the best thing would be to select a different realisation that has zero internally generated trend.

We checked the internally-generated trend in panel a) and it is fairly small over the 150-year period (~7 m/yr), though there are some large decadal trends in both directions as expected in natural variability. In panel a, that we have one glacier advance to a new bump, one retreat to a new bump, and the rest not transitioning (still fairly tightly clustered), suggests to us that this isn't a terribly biased realization. It may be fairly good for illustrating the natural state - one might expect to see some sustained advances or retreats in a population, depending on local geometry. Similarly, approximately 90% of glaciers retreat in panel (b), which is indeed somewhat above the median in panel (c) but not an outlier, so we think this is a reasonable realization to show.

However, we like the idea of adding the internally-generated trend line for full transparency, and will add that to the revised figure:

[Figure]

Section 5 General: As with section 4, what statistical testing would be required to demonstrate that a population of glaciers was retreating under climatic forcing, relative to a population fluctuating with no climatic trend, at a given confidence level?

The tests for sampling error that we mention above could be considered if an aleatory ensemble were run for the population of glaciers, but as noted above, it is worth distinguishing sampling issues that could be ameliorated by running larger ensembles, vs. uncertainties in model parameters and other assumptions that are baked into the ensembles. We expect that these epistemic uncertainties are likely to be a larger barrier for overall confidence. These issues might require statistical testing themselves, but would likely depend on the particular case. This is related to an overall comment from the other reviewer (see below), regarding the sensitivity of attribution assessments to our confidence in the various assumptions that must go into them. We've tried to shore up statements that a range of assumptions will need to be tested, and we expect this would apply to the population-based attribution framework too.

Section 5 General: As a closely related point to the one above, I found myself wondering what is wrong with just asking what fraction of glaciers in an area have retreated in the real world. If enough Greenland glaciers are monitored, over a long enough time period, any net retreat implies a climatic trend in forcing must be important, does it not? Then the question becomes how many glaciers and how long a time period need to be monitored to provide a given confidence level. This re-states my 'time of emergence' point above. I can't quite link this concept to the work in section 5, but I bet the authors can. (Plus, I bet enough Greenland monitoring data are available to provide a pretty high confidence level.)

We agree with the point that the overall fraction could be a strong metric of change on its own, but we would also stress that for *attribution* of that change, we would still need to compare it with a no-anthropogenic-forcing counterfactual case. We have great observations showing the ubiquity of retreats in recent decades, but for attribution we still to understand how much this

goes above and beyond the response to a strong multi-decade trend associated with internal variability; we need a reference point in an unforced world (which typically requires models, in the absence of detailed preindustrial observations). Additionally, a long-term net retreat might still occur if variability pushes a terminus off of a bed peak, which is part of why we think the counterfactual probability or fraction-retreating is so important.

The point about the duration of observations does seem very relevant - and the observations of early retreats on some glaciers in Greenland raise a similar question. This in part motivated our discussion of these early retreats in section 6, and they might provide useful case studies in future work.

General:  Even if the existence of important climatically driven changes in a glacier can be established, that does not imply that the climatic changes are anthropogenic.

We completely agree - assumptions about the anthropogenic forcing are fundamental to attribution. Because attribution conclusions are sensitive to these assumptions, care is needed when inferring the anthropogenic component of a forcing mechanism (e.g., atmospheric or ocean temps). This is indeed why we highlight the difference between early-onset and late-onset trends, and focus a discussion section on the uncertainty in determining the anthropogenic signal in forcing. We did point out that observed climate trends don't necessarily partition the anthropogenic component (2nd paragraph of 6.1), but we will add some further emphasis that attribution of the forcing is a key step:

> *These effects are very clear in the synthetic experiments, where the difference between ensembles**---that is, an anthropogenic climate trend---**is simply imposed. **However, this trend must ultimately be inferred from observations and models of climate, which is an attribution task of its own.** When targeting real glaciers, it will be important to evaluate assumptions about the onset **and magnitude** of **anthropogenic** trends built into the model simulations.*

And later:
> *Our results suggest that assessing uncertainty in the  **anthropogenic component** of local climate forcing will be very important for understanding the robustness…*

Conclusions Line 521: natural fluctuations in climatic forcing
Fixed.

---

## Author Comment (AC2)

**Author responses for tc-2021-394, "A probabilistic framework for quantifying the role of anthropogenic climate change in marine-terminating glacier retreats"**

**KEY**
Reviewer comments
Our responses
    *New or adjusted text*

**Reviewer 2**
Summary
The overarching goal of this study is to provide a framework to attribute glacier retreat to anthropogenic climate change.  The authors seek to do this by performing ensemble simulations of glacier retreat in idealized geometries driven by quasi-random climate variability. Overall, I think the study is very well written and illustrated.  The figures were easy to read and interpret and the text provided sufficient motivation and narrative structure to follow the thread. I have some overarching comments, some boring if highly technical comments and some minor comments about wording, but overall my comments are minor.

Thanks very much for the thoughtful and encouraging review. We especially appreciate the overarching comments that get at the fundamental questions that attribution assessments must wrestle with, and it has been a useful process to consider them. Please find our responses below:

Overarching comments.

One of the results of the manuscript is that random climate fluctuations will eventually cause glaciers to retreat.  The time it takes to do so depends on the magnitude of the imposed noise. Of course if glaciers had experienced a stationary stochastic climate in the past this implies glaciers should all be in their retreated position(s) and we wouldn't observe any glaciers in their more advanced positions.  This isn't really a problem for the study because the climate, has not been stationary stochastic and has variability on a range of time scales.  This raises two questions.

Before responding to the two questions below, we also want to make a general comment on the conundrum as to why any glaciers would be found in advanced positions (i.e., still on peaks), as it is indeed a puzzling implication. Your point about the timescales over which climate is stationary is a good one, and other slow processes might also play in on long timescales, such as sedimentation and isostatic effects (as we note in section 6.2). An additional consideration is that since subglacial/proglacial topography tends to have many bumps, it's possible that glaciers could have previously retreated from different topographic highs and wound up on others. The attribution question is just going to be framed around retreat from whatever the glacier's pre-retreat position was. Admittedly, this is obscured by our single-peak idealized

geometry, but we can see this in the simulations with random bed topography. The initialization period includes multiple peak-to-peak retreats for many glaciers, but the attribution experiment is focused on where they happen to be after 5000 years. While it's not meant to be an actual reconstruction of the last 5000 years, hopefully it adds some insight to the problem.

1.  As the authors point out, modern glaciers are responding to both past and present climate forcing.  We know that glaciers have advanced during colder periods.  For example, some glaciers may have advanced during the Little Ice Age and then retreated and, depending on glacier size, glacier response to the Little Ice Age and other climate anomalies would overlap with the anthropogenic climate interval.  However, the model clearly shows rapid retreat with little advance.  This raises the question of whether the model (or perhaps geometry?) is capable of simulating prior glacier advance.  If advance is not possible, then it seems possible that the model is overestimating the probability of glacier retreat in response to climate forcing (?), potentially biasing the statistical inference.  To put this another way, in the authors model the glaciers will eventually retreat irrespective of anthropogenic climate forcing and the only thing that warming does is increase the probability that this occurs sooner.  In this scenario, anthropogenic climate change only affects rates and not states (i.e., the time of retreat can be accelerated by warming, but retreat is ultimately going to happen irrespective).  It would be more satisfying intellectually if the authors could turn around and also attribute glacier advance to periods when the climate was colder, like we have observed in the historical and paleo-records.

First, on the question of advance: For the single-peak simulations, the fact that we only see rapid retreats and not advances is a consequence focusing specifically on a glacier initially near a bed peak (combined with the hysteresis of marine-ice-sheet instability). So we'd agree that this reflects a bias towards retreat, but this is (or was) the state of many real glaciers prior to retreat. Ultimately this is the unique characteristic that we are trying to address with this framework - it's fundamentally conditioned on starting relatively near a peak. We chose the simple single-peak case as a clear way to explore this in the context of attribution, though we agree that it doesn't address how the glacier got there and this is also an important question. Indeed, we note this in the discussion section on initial conditions (last paragraph of 6.2).

The model can indeed exhibit unstable advance on retrograde slopes, though this situation would require a mean state near the bottom of a trough (for variability to trigger advance), or a nonstationary climate forcing to push it into such a region (as you suggest). The single-peak geometry isn't ideal for illustrating this, but we can explore this with our group of random bed geometries. Below, we show a group of glaciers on different bumpy topographies subject to a positive step change in surface mass balance (and the usual random frontal ablation variability). The glaciers do advance, and occasionally hit reverse slopes and undergo rapid unstable advance. We'll add this figure to the supplement.

[Figure]

*Figure SN: Response of 50 glaciers to a 30% increase in surface mass balance (along with stationary variability in frontal ablation). Each glacier has unique and random bed topographies, as described in the main text. Termini advance through the bumpy topography, including some cases with large rapid advances when they encounter retrograde slopes.*

We also agree with the assessment that a non-zero probability of retreat without a trend implies that retreat would eventually occur at some point. Focusing on changes in the rate (or probability per time) is intentional - we think this is a useful way to think about observed retreats whent threshold processes are at play, since the magnitude of response after breaching thresholds (at least thus far) depends so strongly on non-climatic factors such as geometry. It can seem a bit odd that our counterfactual case implies inevitable retreat at some point, but we'd argue this implication is currently what is on the table, given the lack of attribution assessments for observed marine-terminating glacier retreats. The magnitude of the long-term response is of course a different situation under which to assess the role of anthropogenic forcing.

2. The authors break the probability distributions into a component related to (random) natural variability of the climate forcing and a component related to parameter uncertainty. This is fairly standard, at least in the glaciological literature and it follows from numerous studies in engineering. However, it makes a potentially large assumption: that we understand the system well enough that the model uncertainty largely derives from a handful of parameters that are imperfectly known. There is another possibility that also has to be considered which is that the underlying parameterizations are either not complete or fail in different climate scenarios. This seems especially relevant when dealing with submarine melt, iceberg calving, shear margin weakening, subglacial hydrology, etc, none of which are especially well understood. To be clear, my understanding is that the entire formalism presented here can be applied to any model irrespective of the models fidelity. For example, ca 2000 one could apply this same

method using Shallow Ice Approximation models that don't account for longitudinal stresses or marine ice sheet instability.  These models would require much more oomph from the climate variability to drive retreat because they lack crucial physics.  But the same formalism would allow "attribution".  Hence, a crucial point made by Shepherd (2021) is that we also have to consider all the alternative hypotheses that could also account for the observations.  Shepherd (2021) described how to do this using Bayesian analysis through the use of the "complement".  The trick is that one can formally include how much confidence we have in the model vs alternative models/explanations.  I don't propose that the authors utilize this approach here, but I would like to make sure that they are aware of it and urge them to consider the possibility that their model might not be as physically robust as one might assume from the discussion in the text.  I will note that this is gently hinted at near line 215, but it does seem important to emphasize that the attribution is very sensitive to model assumptions and ultimately, this effect needs to be quantified.

 These are great insights and we agree wholeheartedly that attribution is always conditioned on model assumptions, and the confidence therein will need to be assessed. Our analysis of the different trend scenarios follows in the same spirit: we will probably have to test over a range of assumptions about the anthropogenic climate forcing. We did try to emphasize that attribution is contingent on such assumptions, and they therefore need to be clearly stated (see last sentences in sections 6.1 and 6.3). That said, you raise a good point that this also extends to incomplete model physics, or other processes that could explain retreats. And, we appreciate being pointed to the Shepherd (2021) paper, which contains a number of points relevant to designing these experiments. We think the statements in 6.1 and 6.3 play a part in conveying this overall point, but we will also revise the "outlook" part of section 7 to round it out more broadly.

> *As discussed in the previous section, uncertainties in a glacier's preindustrial position and in the onset of anthropogenic forcing pose fundamental challenges for attribution, **as do uncertainties in key physical processes such as calving, submarine melt, and glacier sliding**. Despite these gaps, our view is that sufficient mechanistic understanding and observational constraints exist to motivate ensemble-based attribution assessments on well-observed glaciers. **In light of these uncertainties, it may be necessary to test over a range of plausible assumptions about glaciological processes and climate forcing that are consistent with observed retreats. An overall assessment would ideally combine results from this range according to our confidence in each assumption (e.g., Shepherd, 2021).***

Technical comments:

1.  How do the authors define noise and what does it mean for the forcing to be ``random''?  My understanding is that the authors assume mass balance has a secular component with zero-mean fluctuations super-imposed.  But I'm not entirely sure how the fluctuations are defined and

I would encourage the authors to add additional details and equations about how the noise is created in the supplementary materials.  More concretely, I take it that noise is added to the surface mass balance?  For a zero-mean Gaussian process, the noise is not smooth and differentiable, so we would then need to integrate it in the form:
$h(t+\Delta t) = h(t) + \Delta t (f(t)+S(t)) + \sqrt(\Delta t) \sigma(t,h)$
where \sigma(t,h) is the standard deviation of the Gaussian process and S(t) is the secular component and I have defined f(t) as the divergence term in the mass balance  (or other terms in the equation).  Note that the random noise term is multiplied by the square root of the time step in a Brownian process.  There is a literature on integrating stochastic differential equations using colored (as opposed to white) noise, but that far exceeds my mathematical acumen.   It would be helpful to me to see more details summarizing how the noise is created and how the stochastic differential equation is then integrated along with demonstrations of numerical convergence using both varying time step size and grid resolution.  I don't request a host of convergence studies added to the paper, but a few sentences explaining that they were done and the results of the convergence experiences.

We generate random climate anomalies with the prescribed statistics outside of the model, and then read them in at each timestep, averaging as necessary for timesteps longer than the sampling interval of the timeseries (which should be consistent with your point about scaling the standard deviation by sqrt(Δt)). We'll clarify this procedure where the noise is introduced in section 2:

> *We generate **timeseries of annual climate anomalies (zero-mean)** using a Fourier transform method (see Percival et al., 2001; Roe and Baker, 2016; Christian et al., 2020). … **At each model time step, we add the corresponding anomaly to the mean SMB or frontal ablation term in the model's continuity equation, averaging over the time step if it is greater than 1 year. For most simulations, we use a model time step of 5 years; we found little effect on terminus fluctuations with time steps of 1--10 years, as the high frequencies of climate variability are strongly damped by the ice dynamics either way (supplemental fig. SN).***

And we will the following figure to the supplement showing the effect of a longer time steps, along with equations for generating the AR-1 noise.

[Figure]

2. The frontal ablation parameterization is intriguing, but I have some questions and comments about this. As I understand it, this approach involves applying a large, negative surface mass balance localized at the last grid point at the grounding line and labeling this a "flux" or frontal ablation term. This seems intuitive at first: the flux term is removing ice at the terminus. The large frontal ablation causes a surface slope between the last two grid points in the (discretized) model. That this is in fact a surface ablation parameter can be seen by moving the frontal ablation to the right hand side of the ice thickness equation. For example, writing the flux as q, an upwind finite difference has the form:

$h(x,t+\Delta t) = h(x,t) + \frac{q(x) - q(x-\Delta x)}{\Delta x}\Delta t + \frac{h \dot m}{\Delta x}\Delta t + S(t)\Delta t$

Note that the second to last term is the frontal ablation term. In the limit that $\Delta x$ becomes small, the surface ablation term becomes large, leading to an increased effective surface mass balance at that point. I have messed around with this type of parameterization a lot in the past (e.g., Bassis et al., 2017) and could not convince it to converge numerically under grid refinement. Instead, this type of parameterization created an unphysical singularity in the slope/thickness of the glacier that became larger and larger as the grid spacing became finer and finer. To cure the singularity, I had to regularize the frontal ablation term, recognizing that the surface ablation needs to be spread out over a characteristic length scale (I used 1 ice thickness). Doing this cured the lack of convergence and provided more physical surface slopes when using small grid spacing. But the results will depend modestly on the regularization scheme. Because of my experience, I would recommend considering a numerical convergence study to assess if the results are independent of grid resolution, time stepping, etc. This is not to say that the authors scheme is problematic, but it would be reassuring to provide some additional tests. To be honest, the entire attribution framework would still work even if the model does depend on the grid spacing. It would just emphasize my previous point that a real attribution requires some estimate of our confidence in model physics and numerics.

Bassis, J., Petersen, S. & Mac Cathles, L. Heinrich events triggered by ocean forcing and modulated by isostatic adjustment.Nature 542, 332–334 (2017)

Thanks for bringing this issue to our attention - we hadn't fully considered the dependence on grid size and are glad to be aware of it. We conducted some convergence tests, assessing both the steady state and transient responses. As you found, the slope right at the terminus becomes steeper and eventually the model fails to converge (our limit was around 30 m spacing). It also has an effect on the steady-state position, which would thus imply that the probability of retreat depends somewhat on the grid size. We note that it has no noticeable effect on terminus fluctuations other than a small DC shift corresponding to the change in steady-state position, but as we have shown for other parameters, this can indeed affect the probability of retreat.

Ultimately, we see this as a drawback of this particular frontal ablation implementation, which is in part related to the flotation condition (at least in our model). We agree with your assessment that it doesn't greatly affect the framework presented here (as long as the grid is consistent among simulations), but it is an important effect to note. We will add the following to the description of frontal ablation in section 2:

> *We note that because the frontal ablation is implemented at a single grid point, its effect on surface slopes and the steady-state terminus position depends on the grid size (Supplemental Fig. SN), and also creates numerical convergence issues for very fine grids (< 30 m). However, we use a consistent grid scheme whenever comparing simulations, to this does not affect our overall conclusions.*

And will note it alongside the results on parameter perturbations:
> *We note this may also include choices in numerical methods, such as the manner in which frontal ablation is prescribed (supplemental figure SN)*

Finally, we will present the sensitivity test in the supplement:

[Figure]

*Figure SN. Convergence test for grid size. The frontal ablation term is applied at the last grid point, making its effect on local slopes dependent on grid size. This in turn affects the steady state terminus position. (a) shows steady state profiles near the terminus. (b) plots stead-state terminus against grid size. (c) shows Terminus fluctuations with each grid size. The character of stable terminus fluctuations is not very sensitive to grid size (except for the offset due to different steady-state positions). (d) shows fluctuations with a trend added, which causes retreat for some cases. For the simulation with the coarsest grid size, the offset in terminus position*

*(i.e., (a) and (b)) is enough to prevent retreat in this particular case. This indicates that under this frontal ablation scheme, the probability of retreat could be sensitive to grid size due to its effect on steady-state position, in the same way other parameter perturbations affect the probability of retreat (i.e., Fig. 3 in the main text).*

3.  I think my biggest recommendation is that the authors conduct some numerical convergence studies.  In my opinion, numerical convergence studies are like brushing your teeth: unglamorous, but essential hygiene that needs to be regularly performed to avoid unpleasant surprises.  This is often done and then forgotten about.  Please tell readers what you have done even if you don't show it.

Agreed - See responses above. In short, we have added a supplemental figure for the grid-size convergence issue, and also that the glacier fluctuations don't change appreciably when the model timestep is varied from 1 to 10 years.

Minor comments:

Line 275 and elsewhere: While—>Although.  While is technically supposed to refer to time. Fixed.

Introduction: Grounding line retreat in WAIS maybe related to the MISI (although also ocean forcing!),  which is tied to retrograde bed slope.  But there are other types of retreat.  For example, the disintegration of ice shelves in the Antarctic Peninsula is not tied to bed slope (because the ice shelves are freely floating).  Similarly, retreat of Petermann Ice Tongue is also not tied to bed slope.  I think the discussion here is mainly focused on Greenland and grounded glaciers.  This might be something worth emphasizing.

We'll clarify grounded ice early on (2nd paragraph):
    *In both Greenland and Antarctica, **the retreat of grounded ice** has been linked to….*

Bed topography in Figure 1 looks like it is piecewise continuous, but not differentiable.  This can create numerical issues and problems with numerical convergence in models that assume the ice thickness is smooth and differentiable.

Correct, the idealized bed peak is mathematically sharp. This can indeed create issues with this model if the grid spacing is too coarse as the stretched grid aliases the topography as the grounding line varies. However, for our grid spacing on the order of 100 m (near the terminus), we haven't had any convergence issues even as the terminus retreats over the peak - likely because the bed slopes are relatively low even on the peak, so the vertical errors due to aliasing are only on the order of a few meters.

Line 135: Out of curiosity, why not use a one sided probability distribution (e.g, log-normal) that naturally avoids unphysical adding mass to the terminus?

This seems like a reasonable alternative approach. We had adapted an existing method to generate the Gaussian noise and later found it necessary to truncate negative values, but this would likely work just as well.

Figure 2 makes a key point: In a system that is close to a system with an instability, retreat always occurs and the only question is how long it takes for retreat to initiate.  I don't know that there is strong evidence for this type of behaviors for glaciers.  This might be partially because the climate is not stationary stochastic over this type of time scale.

See response to the overarching comment above, but in short, we agree that the timescale of stationarity is probably part of it, and there may well be other long-timescale processes to consider, such as changes in stability due to erosion, sedimentation, etc..  But ultimately we do see glaciers perched on bed peaks, or having retreated off of bed peaks, which motivates attribution analyses!

Equation (1) in the supplement: I think the exponent is (1/n)-1 and not 1/(n-1).  Please check.
 Thanks for catching this! Fixed.

Equation (4) appears to be missing an ice thickness on the right hand side?  Please check.
 Thanks! Fixed.